# Comparison and Analysis of Radial and Tangential Bending of Softwood and Hardwood at Static and Dynamic Loading

**Vlastimil Borůvka \*** , **David Novák and Přemysl Šedivka**

Department of Wood Processing and Biomaterials, Faculty of Forestry and Wood Sciences, Czech University of Life Sciences Prague, 16500 Prague, Czech Republic; novakdavid@fld.czu.cz (D.N.); sedivka@fld.czu.cz (P.Š.)
\* Correspondence: boruvkav@fld.czu.cz; Tel.: +420-224-383-636

**Abstract:** This paper should primarily lead to a targeted expansion of the database dealing with bending characteristics, and thus help to understand the static and dynamic bending strength depending on the direction of external forces. Wood is very often used in the structural elements of buildings and wood products (e.g., furniture), in which there is both a static load, and in many cases a dynamic load, whilst the direction of loading is usually not considered. Specifically, the paper focuses on determining the bending strength and impact strength of seven economically-important wood species in the Czech Republic. The research includes not only the above-mentioned strength characteristics, but also the elastic characteristics, i.e., the static modulus of elasticity, and the dynamic modules of elasticity determined using the ultrasound and resonance methods. The procedure was methodologically in accordance with the valid harmonized standards or the usual methodological regulations. The most significant finding can be considered that the largest difference of the mean values of impact strength in the radial direction to the tangential direction was recorded for spruce wood, namely 50.3%. Slightly smaller differences were observed for larch wood, i.e., 41.2%. Minor differences of around 20% were recorded for beech, ash and oak wood. A difference with the opposite trend was recorded for birch wood rather than for the above-mentioned woods, namely −9.5%. Linden wood showed almost no difference (−0.8%). With regard to static bending strength, it was found that the largest difference (radial/tangential) was recorded for oak wood, i.e., 7.9%, while smaller differences were found for linden wood amounting to 6.6% and birch 4.7%. For spruce, larch, beech and ash wood, these differences are negligible. Another finding is that the dynamic modules of elasticity are greatly overestimated compared to static modules of elasticity. In the case of the examined wood of coniferous trees, these differences were up to a maximum of 20%. For wood of wood species with a diffuse-porous structure of wood, the differences were more pronounced, i.e., the range of 36% to 68%, and for wood species with a ring-porous structure in the range of 21% to 43%.

**Keywords:** wood; bending strength; impact bending strength; modulus of elasticity; density; ultrasound; resonance; anisotropy; variability of properties

---

## 1. Introduction

As is generally known, the values of static and dynamic bending strength of wood change depending on the direction of external forces (radial and tangential loading direction). In the case of static strength, the difference between the directions is by no means significant, but in the case of dynamic strength, it certainly is. Several foreign authors [1–5] have already dealt with dynamic bending strength, but none of them dealt with changes in values depending on the direction of external forces. This issue has essentially not been dealt with in any way for the wood of wood species commonly

found in the Czech Republic. Only Požgaj et al. [6] have addressed this issue in the Central European region, but even in this case, the database of these characteristics is very limited. Wood is often used in structural elements in which both static loading and, in many cases, dynamic loading occur [7], with the direction of loading usually not being taken into consideration [8–11].

Static bending strength is expressed by the relationship between the bending moment and the cross-sectional modulus, and the dynamic bending strength is expressed by the degree of work consumed to rupture the body [1]. The influence of time on the resulting bending strength values is among the decisive factors [2,12], as well as the type of wood [1], the direction of loading [13], moisture content, deflection of wood fibers and the occurrence of defects [14]. Deflection of wood fibers and the occurrence of defects affect the speed of sound propagation in wood [15,16]. Since sound propagation is dependent on the environment in which it propagates [17], this knowledge can be used in nondestructive methods of material testing [18,19]. The ultrasound and resonance method was used in this work to detect potential defects and errors in wood (deflection of wood fibers), to partially eliminate wood heterogeneity and, as a result of this, to ensure the most ideal "homogeneity" of parallel test specimens. Verification of the parallelism of the test specimens guaranteed more representative values determining some of the elastic and strength characteristics. From the obtained values of the time of passage of the wave via the ultrasound and resonance method, it is possible to determine dynamic modules of elasticity [20,21], which were also determined and evaluated in this paper. The static modulus of elasticity obtained during the static bending strength test was also assessed.

This paper should primarily lead to a targeted expansion of the database dealing with this issue and thus help to understand static and dynamic bending strength depending on the direction of external forces. This paper does not aim to deal with bending characteristics from the point of view of long-term load, for which different specifics apply than in the case of short-term static or dynamic bending load.

## 2. Materials and Methods

### 2.1. Preparation and Conditioning of Test Specimens

The production of test specimens was based on standard ČSN 49 0101 [22]. The samples were made of coniferous and deciduous wood species. Cut-outs from the trunks were made in the stands of the school forest enterprise of the Czech University of Life Sciences in Kostelec nad Černými Lesy. Selected from the coniferous trees were the Norway spruce (*Picea abies* L.), i.e., wood species with mature wood, and European larch (*Larix decidua* Mill.), wood species with a core. Two representatives with a ring-porous wood structure and three representatives with a diffuse-porous wood structure were selected from deciduous wood species. In wood species with a ring-porous wood structure, the common oak (*Quercus robur* L.) and ash (*Fraxinus excelsior* L.) were included. The European beech (*Fagus sylvatica* L.), the small-leaved linden (*Tilia cordata* Mill.) and the silver birch (*Betula pendula* Roth) were included in wood species with a diffuse-porous wood structure. Central boards were manipulated from the cut-outs of these seven wood species which, after drying to an air-dry state (approximately 15% moisture content), were then subjected to further processing. Test specimens measuring $20 \times 20 \times 300$ mm were made from individual boards. A total of 1120 test specimens were made, i.e., 160 pieces from each type of wood. During production, emphasis was placed on the integrity (purity) of samples and their continuity (4 parallel samples in a row), which was monitored mainly in the individual static and dynamic loading experiments. The principle of dividing the board up to the final test specimens is shown in Figure 1.

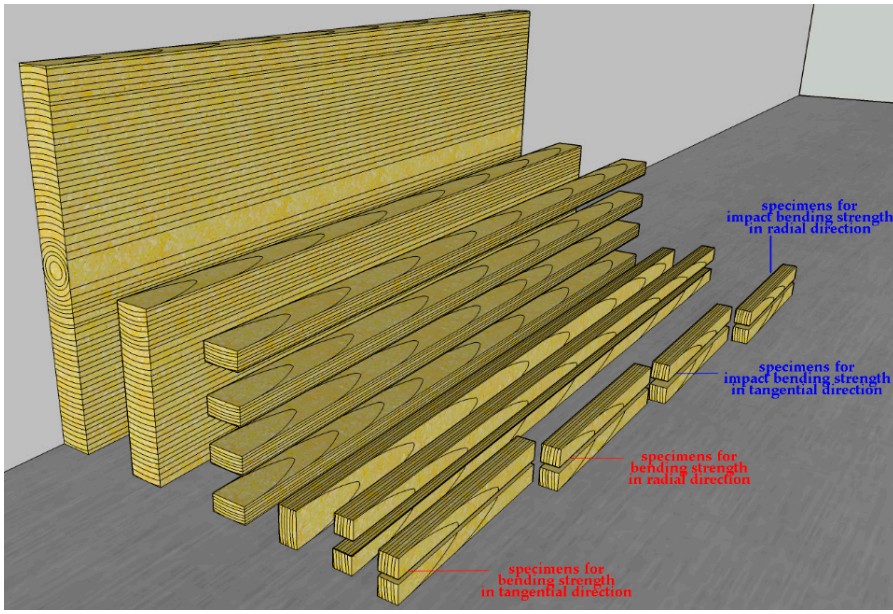

**Figure 1.** Scheme of preparation of test specimens from the central board.

The moisture content of the prepared samples was subsequently adjusted by means of an air conditioning chamber ClimeEvent C/2000/40/3 (Weiss Umwelttechnik GmbH, Reiskirchen, Germany). The basic parameters in the air conditioning chamber were set, so that the resulting absolute moisture of the wood was about 12%. Therefore, the air-conditioning of the test specimens took place until the equilibrium moisture content was stabilized in a controlled environment with an air temperature of 20 ± 2 °C and a relative humidity of 65% ± 5%. This process guaranteed roughly the same moisture conditions for all of the test specimens from each wood species.

*2.2. Determination of Density and Moisture Content*

Density was determined for the air-conditioned test specimens according to the procedure specified in standard ČSN 49 0108 [23]. Individual specimens were weighed using laboratory scale Kern PCB 2500-2 (KERN & SOHN GmbH, Balingen, Germany) with an accuracy of 0.01 g, and the dimensions of the test specimens were measured using caliper Kinex 6040-27-150 (KINEX Measuring s.r.o., Prague, Czech Republic) with an accuracy of 0.01 mm. The following general formula was used to calculate density $\rho_w$ (kg/m$^3$):

$$\rho_w = \frac{m_w}{V_w},\tag{1}$$

where $m_w$ (kg) is the weight of the wood at absolute moisture $w$ (%) and $V_w$ (m$^3$) is the volume of wood at absolute moisture content $w$ (%); the calculation of which was based on standard ČSN 49 0103 [24] according to the following formula:

$$w = \frac{m_w - m_0}{m_0} \cdot 100,\tag{2}$$

where $m_w$ (kg) is the weight of the moist wood and $m_0$ (kg) is the weight of the absolutely dry wood. The samples were dried to zero moisture content in dryer Binder FD 115 (Binder Inc., Tuttlingen, Germany) at a temperature of 103 ± 2 °C.

*2.3. Experiments as Part of Ultrasound and Resonance Methods*

In order to detect and eliminate errors in wood and verify the relative "homogeneity" of the test specimens, or approximately similar heterogeneity, nondestructive methods based on sound propagation in wood were used. For each of the methods, the time (μs) required for the sound wave to

pass through the test specimen was obtained. The obtained times were compared to each other for individual test specimens, which followed up each other within the test. Dynamic modules of elasticity were then calculated from the obtained times. Two ultrasound methods and one resonance method were used to obtain the times. A test apparatus FAKOPP Ultrasound Timer UT-06/2013 (Fakopp Enterprise Bt., Ágfalva, Hungary) was used for the ultrasound method (see Figure 2) and a manual [25]. This apparatus is equipped with two removable piezoelectric sensors. Two types of probes were used to determine the passage of time differing in shape and structure (wedge, square) depending on the applied method, i.e., which method was used to achieve the required quantities.

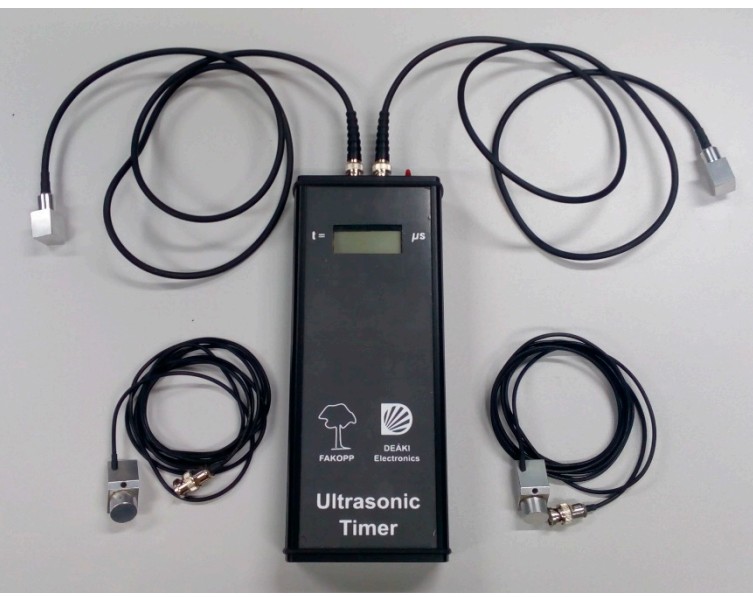

**Figure 2.** Photo of equipment Ultrasonic Fakopp Timer.

The first method consisted of applying wedge probes to the surface (radial, tangential) of the test specimen at a predetermined distance. The measurements were performed at distances of 60, 100, 140, 180 and 220 mm from each other, but time from a distance of only 140 mm was used to compare parallelism. Other times were used for time correction at a zero probe distance and subsequent calculation of the dynamic modulus of elasticity, which can be determined by the following relationship:

$$MOE_{dyn} = c^2 \cdot \rho, \tag{3}$$

where $MOE_{dyn}$ (Pa) is the dynamic modulus of elasticity, $c$ (m/s) is the speed of sound and $\rho$ (kg/m$^3$) is the wood density.

The second method consisted of applying square piezoelectric sensors to the faces of the test specimens, i.e., cross sections. In this case, the distance of the probes from each other corresponded to the total length of the specimen. Even in this method, the dynamic modulus of elasticity can be calculated on the basis of the obtained quantities, but the correction is determined by placing the probes together and subtracting them from the resulting measured value.

The resonance method consisted of determining the frequency, i.e., the number of repetitions of periodic oscillations per unit time (Hz). The sample placed on the rubber pads was initiated by an impact to the front by a quick blow of a steel hammer, which had a weight in the range 0.5%–5% of the weight of the tested sample. An ECM 8000 microphone (Behringer Spezielle Studiotechnik GmbH, Willich, Germany), which picked up the sound of the blow, was installed at the second front of the sample. This obtained signal was subsequently transferred for processing to a UR22 MK2 amplifier (Steinberg GmbH, Hamburg, Germany); see Figure 3a. The amplifier then sent the already-modified (amplified) signal to the computer for further processing. The signal was then processed in the PC

using FFT (Fast Fourier Transform) software analyzer (Fakopp Enterprise Bt., Ágfalva, Hungary)—see Figure 3b, which, after setting the necessary parameters (sensitivity, frequency range, etc.), processed the sent data to the resulting frequency (Hz). The resulting frequency had to be within ± 20% of the expected frequency, which was calculated according to the following formula:

$$f = \frac{2500}{l},\qquad(4)$$

where $f$ (Hz) is the expected frequency and $l$ (m) the length of the test specimen. The obtained values (Hz) were subjected to a recalculation for the phase velocity $c$ (m/s) of wave propagation according to the following formula:

$$c = 2 \cdot l \cdot f,\qquad(5)$$

where $l$ (m) is the length of the test specimen and $f$ (Hz) is the measured frequency. From the obtained velocity, the dynamic modulus of elasticity was calculated according to the above Formula (3). The following formula was used to recalculate the time $t$ (μs) needed to compare "homogeneity":

$$t = \frac{l}{c},\qquad(6)$$

where $l$ (m) is the length of the specimen and $c$ (m/s) is the phase velocity of the wave propagation.

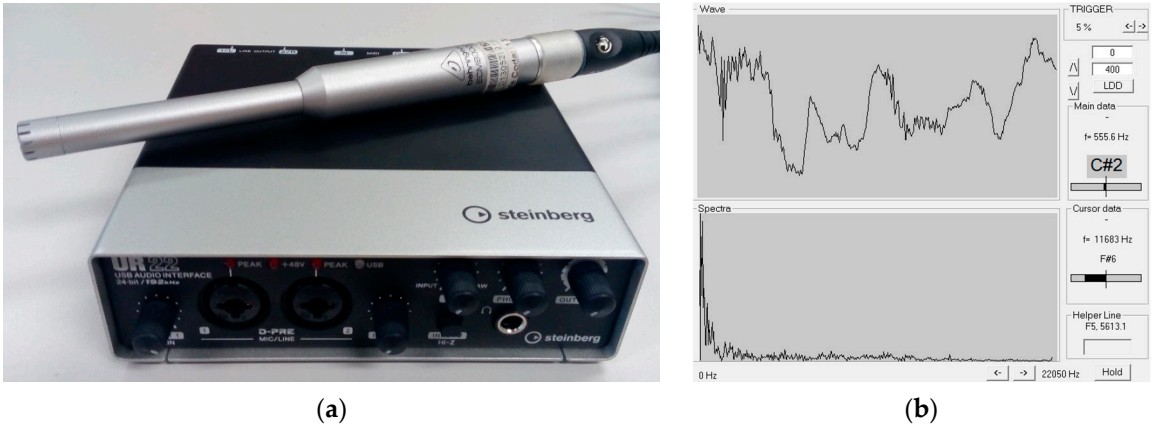

(**a**)　　　　　　　　　　　　　　　　　　　(**b**)

**Figure 3.** Photos of microphone ECM 8000 and amplifier UR22 MK2 (**a**) and record of the graphical interface of software of FFT analyzer (**b**).

*2.4. Experimental Determination of Static and Dynamic Bending Characteristics*

The determination of static bending strength (modulus of rupture) was based on standard ČSN 49 0115 [26]. The determination of the static modulus of elasticity was based on a combination of standards ČSN EN 310 [27] and ČSN 49 0116 [28]. A Tira 50 kN test machine (Tira GmbH, Schalkau, Germany) was used to obtain the required values. A test specimen with a rectangular shape measuring 20 × 20 × 300 mm was placed on two supports with a diameter of 30 mm and at a distance of 240 mm from their centers. With respect to the direction of loading, the force in the radial or tangential direction, transmitted by the load head with a constant speed of displacement, began to act on the test sample stored in this way. The loading rate of the material had to be set so that the specimens raptured in 1.5 min with a deviation of ± 0.5 min. The maximum force value was determined using a force sensor and the TIRA-test software. The following formula was used to calculate the static bending strength values $\sigma_{max}$ (MPa):

$$\sigma_{max} = \frac{3 \cdot F \cdot l_0}{2 \cdot b \cdot h^2},\qquad(7)$$

where $F$ (N) is the maximum load force, $l_0$ (mm) is the distance between the centers of the supports, $b$ (mm) is the width and $h$ (mm) is the height of the specimen.

The static modulus of elasticity is based on the stress induced to the test specimen by the bending moment, i.e., from the relationship below the limit of proportionality, where the stress is directly proportional to the relative elongation. The values are obtained from the linear part of the (working) load curve from the range of 10% to 40% of the maximum load values. The following formula was used to calculate the static modulus of elasticity $MOE_{stat}$ (MPa):

$$MOE_{stat} = \frac{l_0^3 \cdot \Delta F}{4 \cdot b \cdot h^3 \cdot \Delta y}, \tag{8}$$

where $l_0$ (mm) is the distance between the centers of the supports, $b$ (mm) is the width and $h$ (mm) is the height of the test specimen; $\Delta F$ (N) is the difference between the forces and $\Delta y$ (mm) is the difference in deflections.

The determination of impact strength (toughness) was based on standard ČSN 49 0117 [29]. A Charpy hammer (CULS, Prague, Czech Republic) was used to perform the test. This test consisted of placing the test specimen on supports, taking into account the direction of loading, which was examined within the individual wood species. The distance between the centers of the supports was 240 mm. Then the test specimen stored in this way was ruptured by a hammer blow. The impact bending strength $A_w$ (J/cm$^2$) was determined using the following formula:

$$A_w = \frac{W}{b \cdot h}, \tag{9}$$

where $W$ (J) is the work consumed to break the specimen, $b$ (cm) is the width and $h$ (cm) is the height of the specimen.

### 2.5. Image Analyses

Specimens measuring $1 \times 1 \times 1$ cm were manipulated from the test samples with a chisel and a saw. The specimens thus prepared were immersed in a container with water until the wood softened. After reaching the desired state, the individual bodies were fixed in a GSL-1 microtome (Fritz Schweingruber, Birmendorf, Switzerland), so that the cutting direction corresponded to the requirement for the final preparations, and they were subsequently coated with starch for a better cut [30]. The method specified, for example, by Bär et al. [31] or Buchwal et al. [32], was then used. Upon completion of the preparations, images were taken with a Nikon D7100 digital camera (Nicon corp., Minato, Japan) mounted on a Nikon Eclipse 80i microscope (Nicon corp., Minato, Japan) to achieve the desired magnification.

Samples from moisture determination measuring $20 \times 20 \times 30$ mm, which were created by cutting the ends from 300 mm long samples, were used to determine the width of the annual rings. An image recording at a resolution of 800 DPI was taken from the front surfaces of the samples using a scanner. This was followed by the measurement of individual annual rings using software program NIS Elements AR (Laboratory Imaging s.r.o., Prague, Czech Republic).

High-speed camera FASCAM Mini UX 50 (IMAGICA DIGIX Inc., Tokio, Japan) was used to record the course of tests under static and dynamic loading, on which a frame rate of 3200 frames per second was set. The image was processed using a PC and Photron FATCAM Viewer 3 (IMAGICA DIGIX Inc., Tokyo, Japan) software program. The software program made it possible to determine the time period from the loading of the specimen to its rupture, which logically applied in particular to dynamic load experiments, where this process is monitored within milliseconds. Another reason was to monitor the formation of cracks.

*2.6. Data Evaluation*

The data and results of the investigated properties were processed in graphical and tabular form using program STATISTICA Version 13.4.0.14 (TIBCO Software Inc., Palo Alto, CA, USA). In this program, basic descriptive statistics and multifactor analysis ANOVA were used, which served to capture the trend of the investigated properties. Duncan's test was also used for multiple comparisons of some of the evaluated properties depending on the direction of external forces. Correlation matrices were used to express the dependences of individual investigated properties. A uniform significance level of $\alpha = 0.05$ was used for all statistical analyses.

## 3. Results and Discussion

The basic descriptive statistics of all monitored quantities for woods of individual wood species are specified in Tables 1–7. Changes in the values of bending strength, impact strength and static and dynamic modulus of elasticity depending on the direction of loading are shown in Table 8. In this table, the changes are given in percentage expression, where a change was monitored in the values of individual detected quantities in the radial direction against the values in the tangential direction. The following are detailed statements on the individual quantities, as well as graphical visualization or correlation dependencies between quantities.

**Table 1.** Basic Statistical Analyses of the Properties for Spruce Wood.

| Properties | Load Direction | Minimum | Mean | Maximum | Std.Dev. | Coef.Var. (%) |
|---|---|---|---|---|---|---|
| Modulus of Rupture (MPa) | Radial | 64.3 | 74.6 | 84.4 | 4.8 | 6.5 |
| | Tangential | 40.6 | 75.5 | 92.0 | 9.7 | 12.8 |
| Toughness ($J/cm^2$) | Radial | 4.9 | 9.8 | 18.2 | 3.3 | 34.0 |
| | Tangential | 3.7 | 6.5 | 9.1 | 1.2 | 17.8 |
| Static Modulus of Elasticity (MPa) | Radial | 7630 | 8589 | 9658 | 543 | 6.3 |
| | Tangential | 7069 | 8661 | 10,075 | 656 | 7.6 |
| Dynamic Modulus of Elasticity * (MPa) | - | 4570 | 7748 | 11,364 | 1582 | 20.4 |
| Dynamic Modulus of Elasticity ** (MPa) | Radial | 7883 | 10,154 | 13,721 | 972 | 9.6 |
| | Tangential | 8119 | 10,367 | 12,613 | 950 | 9.2 |
| Dynamic Modulus of Elasticity *** (MPa) | - | 7698 | 10,075 | 11,748 | 626 | 6.2 |
| Density ($kg/m^3$) | Radial | 431 | 463 | 522 | 20.2 | 4.4 |
| | Tangential | 424 | 463 | 518 | 20.5 | 4.4 |
| Annual Ring Width (mm) | - | 1.8 | 2.6 | 3.8 | 0.5 | 18.3 |

Valid $N = 40$ for all properties within load direction. * Ultrasound Method (from the front surface—cross section); ** Ultrasound Method (from the side surface); *** Resonance Method; Std.Dev. = Standard Deviation; Coef.Var. = Coefficient of Variation.

**Table 2.** Basic Statistical Analyses of the Properties for Larch Wood.

| Properties | Load Direction | Minimum | Mean | Maximum | Std.Dev. | Coef.Var. (%) |
|---|---|---|---|---|---|---|
| Modulus of Rupture | Radial | 82.9 | 102.5 | 125.1 | 11.3 | 11.0 |
| (MPa) | Tangential | 77.6 | 104.6 | 131.3 | 14.5 | 13.8 |
| Toughness | Radial | 3.0 | 8.1 | 11.6 | 2.4 | 29.9 |
| (J/cm$^2$) | Tangential | 2.2 | 5.7 | 10.4 | 2.3 | 39.6 |
| Static Modulus of Elasticity | Radial | 7900 | 10,358 | 13,132 | 1497 | 14.5 |
| (MPa) | Tangential | 8310 | 11,217 | 13,304 | 1416 | 12.6 |
| Dynamic Modulus of Elasticity * (MPa) | - | 5405 | 11,735 | 15,729 | 2533 | 21.6 |
| Dynamic Modulus of Elasticity ** | Radial | 7193 | 10,280 | 13,159 | 1562 | 15.2 |
| (MPa) | Tangential | 7053 | 11,448 | 14,595 | 1530 | 13.4 |
| Dynamic Modulus of Elasticity *** (MPa) | - | 8469 | 11,641 | 14,700 | 1618 | 13.9 |
| Density | Radial | 622 | 744 | 907 | 80.1 | 10.8 |
| (kg/m$^3$) | Tangential | 634 | 760 | 926 | 88.0 | 11.6 |
| Annual Ring Width (mm) | - | 0.6 | 1.5 | 3.1 | 0.6 | 41.7 |

Valid *N* = 40 for all properties within load direction. * Ultrasound Method (from the front surface—cross section); ** Ultrasound Method (from the side surface); *** Resonance Method; Std.Dev. = Standard Deviation; Coef.Var. = Coefficient of Variation.

**Table 3.** Basic Statistical Analyses of the Properties for Beech Wood.

| Properties | Load Direction | Minimum | Mean | Maximum | Std.Dev. | Coef.Var. (%) |
|---|---|---|---|---|---|---|
| Modulus of Rupture | Radial | 93.5 | 111.6 | 129.9 | 9.4 | 8.4 |
| (MPa) | Tangential | 96.1 | 111.0 | 123.1 | 6.6 | 6.0 |
| Toughness | Radial | 4.3 | 6.1 | 7.4 | 0.7 | 12.0 |
| (J/cm$^2$) | Tangential | 2.8 | 5.2 | 6.9 | 1.1 | 21.2 |
| Static Modulus of Elasticity | Radial | 9271 | 11,479 | 13,516 | 905 | 7.9 |
| (MPa) | Tangential | 9387 | 10,635 | 12,325 | 675 | 6.3 |
| Dynamic Modulus of Elasticity * (MPa) | - | 13,551 | 15,617 | 19,095 | 1037 | 6.6 |
| Dynamic Modulus of Elasticity ** | Radial | 13,424 | 16,070 | 21,339 | 1493 | 9.3 |
| (MPa) | Tangential | 12,299 | 16,483 | 22,329 | 1849 | 11.2 |
| Dynamic Modulus of Elasticity *** (MPa) | - | 12,552 | 15,557 | 18,634 | 1142 | 7.3 |
| Density | Radial | 673 | 727 | 785 | 25.0 | 3.4 |
| (kg/m$^3$) | Tangential | 692 | 731 | 802 | 25.1 | 3.4 |
| Annual Ring Width (mm) | - | 1.6 | 2.8 | 3.7 | 0.4 | 15.2 |

Valid *N* = 40 for all properties within load direction. * Ultrasound Method (from the front surface—cross section); ** Ultrasound Method (from the side surface); *** Resonance Method; Std.Dev. = Standard Deviation; Coef.Var. = Coefficient of Variation.

The wood density of individual wood species was determined after air-conditioning of the test specimens. After the air-conditioning, absolute moisture was ascertained for the individual woods, namely 13.8% for spruce wood, 13.3% for larch, 11.9% for beech, 11.7% for birch, 10.2% for linden, 11.5% for oak and 12.0% for ash. The graph (Figure 4) shows the determined values of wood density for individual examined wood species. The highest average density was recorded for larch wood, i.e., 760 kg/m$^3$ for samples that were tested under loading in the tangential direction. The lowest density was recorded for spruce wood, for which the same average density was found for both sets of samples (radial and tangential direction of loading), i.e., 463 kg/m$^3$. Statistically significant differences in mean values depending on the direction of loading were demonstrated for larch wood, and Duncan's test (Table A3) was used for multiple comparisons. Wood density is highly variable and depends on many factors [5,14]. This claim is evident in the case of larch density, which appears to be nonstandard compared to the values specified, for example, by Tsoumis [5] or Wageführ [33].

Other determined densities of individual wood species coincide with the range of densities reported in the literature [5,33,34], except for oak wood, in which a slightly higher density was found, i.e., 719 or 722 kg/m$^3$.

**Table 4.** Basic Statistical Analyses of the Properties for Birch Wood.

| Properties | Load Direction | Minimum | Mean | Maximum | Std.Dev. | Coef.Var. (%) |
|---|---|---|---|---|---|---|
| Modulus of Rupture | Radial | 71.9 | 112.4 | 132.4 | 12.3 | 10.9 |
| (MPa) | Tangential | 74.3 | 107.4 | 132.4 | 12.7 | 11.8 |
| Toughness | Radial | 4.8 | 9.0 | 12.1 | 1.9 | 21.4 |
| (J/cm$^2$) | Tangential | 5.6 | 10.0 | 12.4 | 1.8 | 18.1 |
| Static Modulus of Elasticity | Radial | 6410 | 11,643 | 13,650 | 1410 | 12.1 |
| (MPa) | Tangential | 9125 | 11,165 | 13,156 | 858 | 7.7 |
| Dynamic Modulus of Elasticity * (MPa) | - | 9872 | 16,883 | 20,295 | 1622 | 9.6 |
| Dynamic Modulus of Elasticity ** | Radial | 10,635 | 18,217 | 30,941 | 2799 | 15.4 |
| (MPa) | Tangential | 13,569 | 18,808 | 28,319 | 2898 | 15.4 |
| Dynamic Modulus of Elasticity *** (MPa) | - | 9300 | 17,074 | 21,082 | 1857 | 10.9 |
| Density | Radial | 624 | 716 | 800 | 29.8 | 4.2 |
| (kg/m$^3$) | Tangential | 616 | 715 | 791 | 32.4 | 4.5 |
| Annual Ring Width (mm) | - | 1.5 | 2.4 | 3.4 | 0.5 | 21.0 |

Valid *N* = 40 for all properties within load direction. * Ultrasound Method (from the front surface—cross section); ** Ultrasound Method (from the side surface); *** Resonance Method; Std.Dev. = Standard Deviation; Coef.Var. = Coefficient of Variation.

**Table 5.** Basic Statistical Analyses of the Properties for Linden Wood.

| Properties | Load Direction | Minimum | Mean | Maximum | Std.Dev. | Coef.Var. (%) |
|---|---|---|---|---|---|---|
| Modulus of Rupture | Radial | 65.7 | 92.8 | 107.6 | 9.3 | 10.0 |
| (MPa) | Tangential | 55.1 | 87.1 | 103.9 | 10.6 | 12.2 |
| Toughness | Radial | 4.7 | 6.5 | 9.1 | 1.0 | 15.8 |
| (J/cm$^2$) | Tangential | 3.8 | 6.5 | 8.3 | 1.0 | 14.9 |
| Static Modulus of Elasticity | Radial | 8246 | 10,061 | 12,028 | 856 | 8.5 |
| (MPa) | Tangential | 6675 | 9332 | 10,952 | 887 | 9.5 |
| Dynamic Modulus of Elasticity * (MPa) | - | 11,643 | 14,276 | 16,833 | 1092 | 7.7 |
| Dynamic Modulus of Elasticity ** | Radial | 11,869 | 15,208 | 19,079 | 1560 | 10.3 |
| (MPa) | Tangential | 10,718 | 15,063 | 19,540 | 1659 | 11.0 |
| Dynamic Modulus of Elasticity *** (MPa) | - | 11,700 | 14,959 | 18,476 | 1345 | 9.0 |
| Density | Radial | 509 | 570 | 638 | 31.9 | 5.6 |
| (kg/m$^3$) | Tangential | 512 | 570 | 639 | 30.0 | 5.3 |
| Annual Ring Width (mm) | - | 1.7 | 2.6 | 4.0 | 0.6 | 23.1 |

Valid *N* = 40 for all properties within load direction. * Ultrasound Method (from the front surface—cross section); ** Ultrasound Method (from the side surface); *** Resonance Method; Std.Dev. = Standard Deviation; Coef.Var. = Coefficient of Variation.

**Table 6.** Basic Statistical Analyses of the Properties for Oak Wood.

| Properties | Load Direction | Minimum | Mean | Maximum | Std.Dev. | Coef.Var. (%) |
|---|---|---|---|---|---|---|
| Modulus of Rupture | Radial | 65.9 | 101.7 | 134.2 | 17.7 | 17.4 |
| (MPa) | Tangential | 65.5 | 94.3 | 131.7 | 17.8 | 18.9 |
| Toughness | Radial | 2.7 | 5.2 | 9.6 | 1.5 | 29.0 |
| (J/cm$^2$) | Tangential | 2.8 | 4.5 | 7.5 | 1.0 | 23.5 |
| Static Modulus of Elasticity | Radial | 7707 | 10,944 | 13,946 | 1901 | 17.4 |
| (MPa) | Tangential | 7307 | 10,127 | 12,895 | 1596 | 15.8 |
| Dynamic Modulus of Elasticity * | - | 9232 | 13,259 | 17,102 | 2020 | 15.2 |
| (MPa) | | | | | | |
| Dynamic Modulus of Elasticity ** | Radial | 8705 | 14,404 | 19,477 | 2378 | 16.5 |
| (MPa) | Tangential | 8163 | 14,261 | 19,543 | 2278 | 16.0 |
| Dynamic Modulus of Elasticity *** | - | 8720 | 13,857 | 18,956 | 2404 | 17.3 |
| (MPa) | | | | | | |
| Density | Radial | 647 | 719 | 794 | 40.7 | 5.7 |
| (kg/m$^3$) | Tangential | 652 | 722 | 811 | 41.5 | 5.7 |
| Annual Ring Width | - | 1.1 | 1.8 | 2.4 | 0.3 | 18.0 |
| (mm) | | | | | | |

Valid *N* = 40 for all properties within load direction. * Ultrasound Method (from the front surface—cross section); ** Ultrasound Method (from the side surface); *** Resonance Method; Std.Dev. = Standard Deviation; Coef.Var. = Coefficient of Variation.

**Table 7.** Basic Statistical Analyses of the Properties for Ash Wood.

| Properties | Load Direction | Minimum | Mean | Maximum | Std.Dev. | Coef.Var. (%) |
|---|---|---|---|---|---|---|
| Modulus of Rupture | Radial | 97.8 | 109.3 | 117.8 | 5.0 | 4.6 |
| (MPa) | Tangential | 100.6 | 110.0 | 117.9 | 3.8 | 3.5 |
| Toughness | Radial | 6.1 | 8.0 | 9.5 | 0.7 | 8.3 |
| (J/cm$^2$) | Tangential | 5.4 | 6.6 | 7.9 | 0.6 | 8.7 |
| Static Modulus of Elasticity | Radial | 10,294 | 11,570 | 12,641 | 599 | 5.2 |
| (MPa) | Tangential | 10,172 | 11,004 | 12,122 | 433 | 3.9 |
| Dynamic Modulus of Elasticity * | - | 12,738 | 14,041 | 16,482 | 701 | 5.0 |
| (MPa) | | | | | | |
| Dynamic Modulus of Elasticity ** | Radial | 11,264 | 16,121 | 22,933 | 1758 | 10.9 |
| (MPa) | Tangential | 12,977 | 15,689 | 19,581 | 1294 | 8.2 |
| Dynamic Modulus of Elasticity *** | - | 13,841 | 15,511 | 19,092 | 886 | 5.7 |
| (MPa) | | | | | | |
| Density | Radial | 611 | 655 | 695 | 12.8 | 2.0 |
| (kg/m$^3$) | Tangential | 609 | 656 | 701 | 13.1 | 2.0 |
| Annual Ring Width | - | 1.3 | 2.9 | 3.6 | 0.5 | 18.4 |
| (mm) | | | | | | |

Valid *N* = 40 for all properties within load direction. * Ultrasound Method (from the front surface—cross section); ** Ultrasound Method (from the side surface); *** Resonance Method; Std.Dev. = Standard Deviation; Coef.Var. = Coefficient of Variation.

**Table 8.** Percentage comparison of a specific quantity (property) in anatomical directions (radial/tangential) for individual specific wood species.

| | Spruce | Larch | Beech | Birch | Linden | Oak | Ash |
|---|---|---|---|---|---|---|---|
| Modulus of Rupture * | −1.2 | −2.0 | 0.5 | 4.7 | 6.6 | 7.9 | −0.6 |
| Toughness ** | 50.3 | 41.2 | 18.1 | −9.5 | −0.8 | 16.3 | 20.6 |
| Static modulus of elasticity | −0.8 | −7.7 | 7.9 | 4.3 | 7.8 | 8.1 | 5.1 |
| Dynamic Modulus of Elasticity *** | −2.1 | −10.2 | −2.5 | −3.1 | 1.0 | 1.0 | 2.8 |

* Bending Strength; ** Impact Bending Strength; *** determined by the Ultrasound Method (from the side surface).

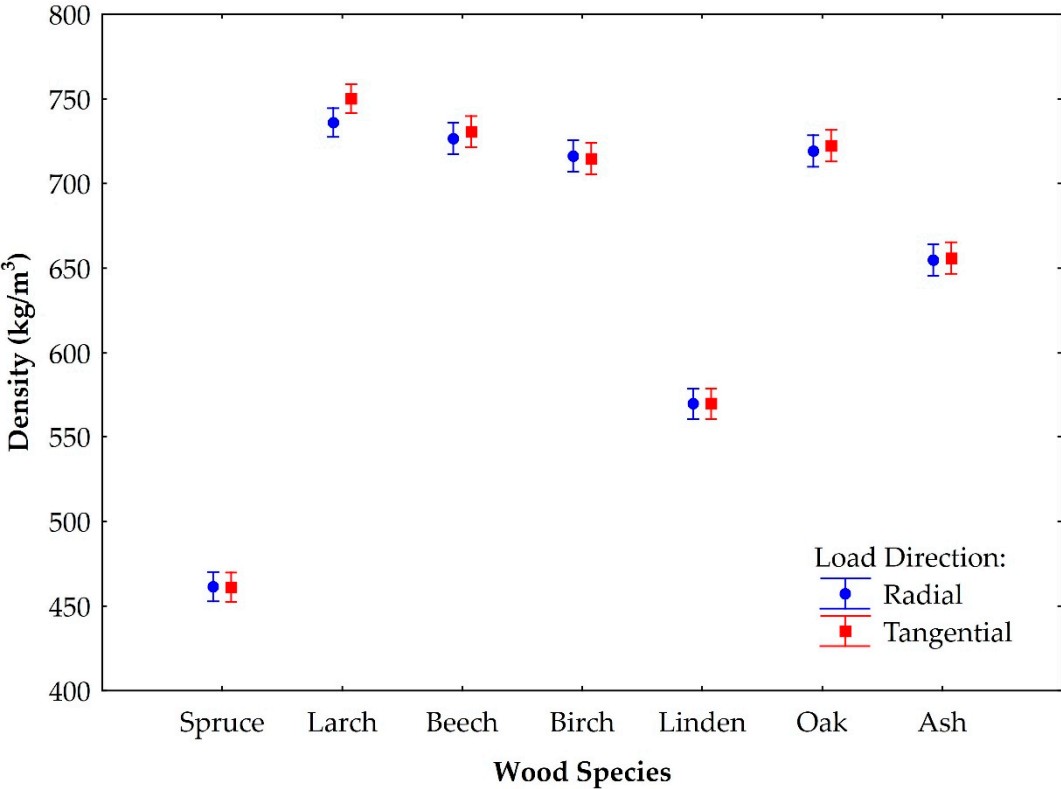

**Figure 4.** Graphic visualization of samples density of specific wood species for determination of the effect of load direction at bending strength and impact bending strength.

Tables 1–7 show the average widths of annual rings for individual wood species. The width and structure of the annual rings is highly variable and influenced by a large number of factors. The width of the annual ring changes both along the diameter of the trunk and along its length [3,5]. The largest width of the annual rings was found for ash wood, i.e., 2.9 mm, and the lowest for larch wood, 1.5 mm, which also showed the highest degree of variability. For birch wood with an average annual ring width of 2.4 mm, a nonstandard factor in the structure of annual rings was found after image analysis. The annual rings were markedly wavy in the cross section (Figure 5d). Figure 5 also shows examples of cross sections of wood of other wood species. All of the images are at 40× magnification.

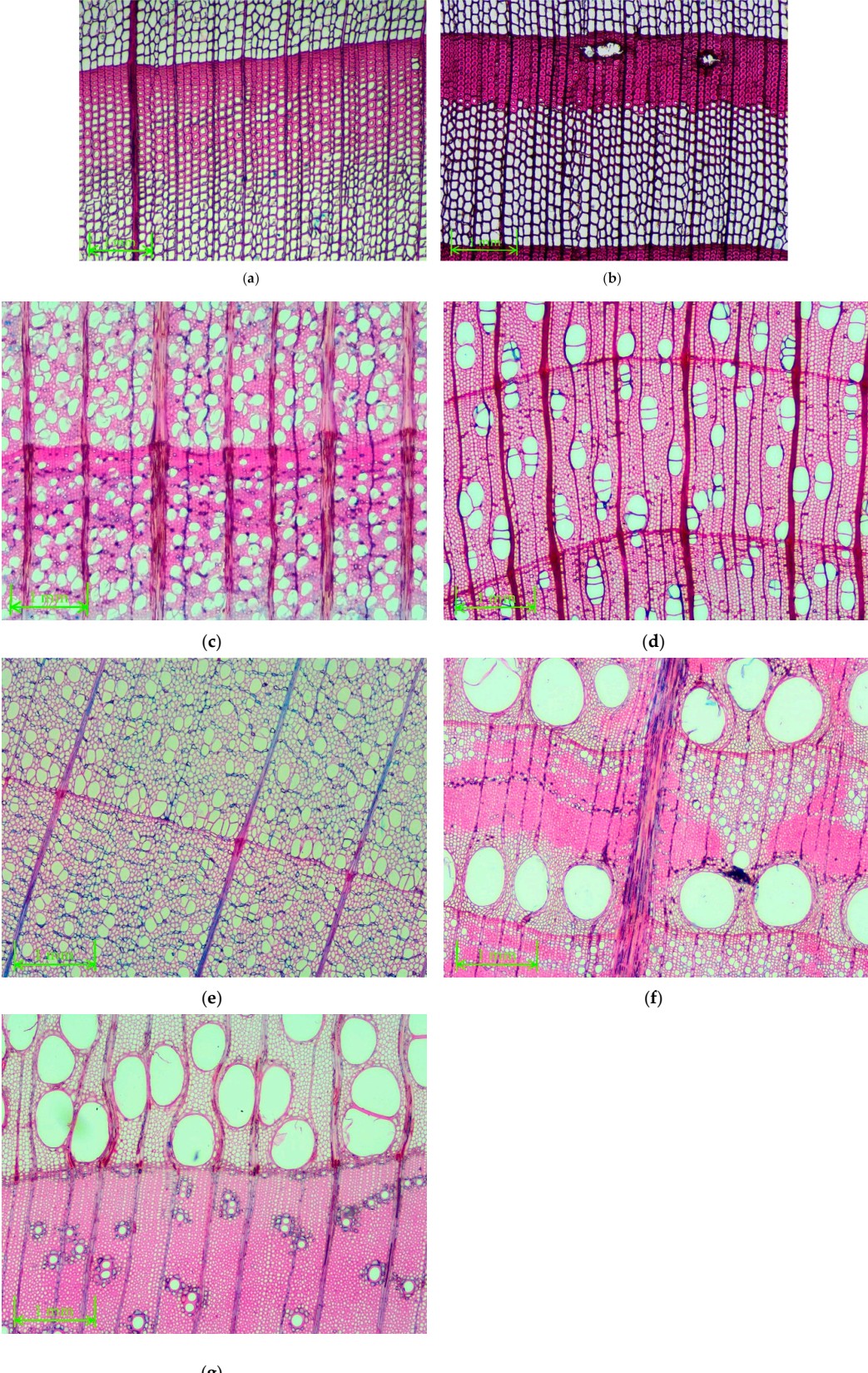

**Figure 5.** Cross section of wood at 40× magnification: spruce (**a**), larch (**b**), beech (**c**), birch (**d**), linden (**e**), oak (**f**) and ash (**g**).

Dinwoodie [2] states in his work that bending strength is highly influenced by the occurrence of defects and the deviation of wood fibers from the longitudinal axis. However, in this work, great emphasis was placed on the quality of the tested material and the relative "homogeneity" of the test specimens was verified, which guaranteed the highest possible representative values of the investigated properties. This homogeneity was verified by the ultrasound and resonance methods. For clarity, the resonance method (Figure 6) is specified, which monitored the time of passage of the sound wave in the test specimens on the basis of the tests they were subjected to. The measured time should be almost identical or similar without significant statistical differences. Only under this assumption were the errors eliminated, and the obtained values determining any of the properties had a more representative weight. After evaluating the data, the differences between the test samples, which followed each other within a specific test, were primarily monitored. Thus, in order to determine the static bending strength, the first two test samples in the lath (1 and 2) were compared, and to determine the dynamic strength (impact strength), the following test specimens, i.e., 3 and 4, were compared. It can be seen from the specified graph that this assumption was fulfilled, and the samples did not show any significant differences amongst themselves. A significant difference was observed for birch wood between samples 1 and 2, i.e., for wood samples in which annual ring corrugation was observed (Figure 5d).

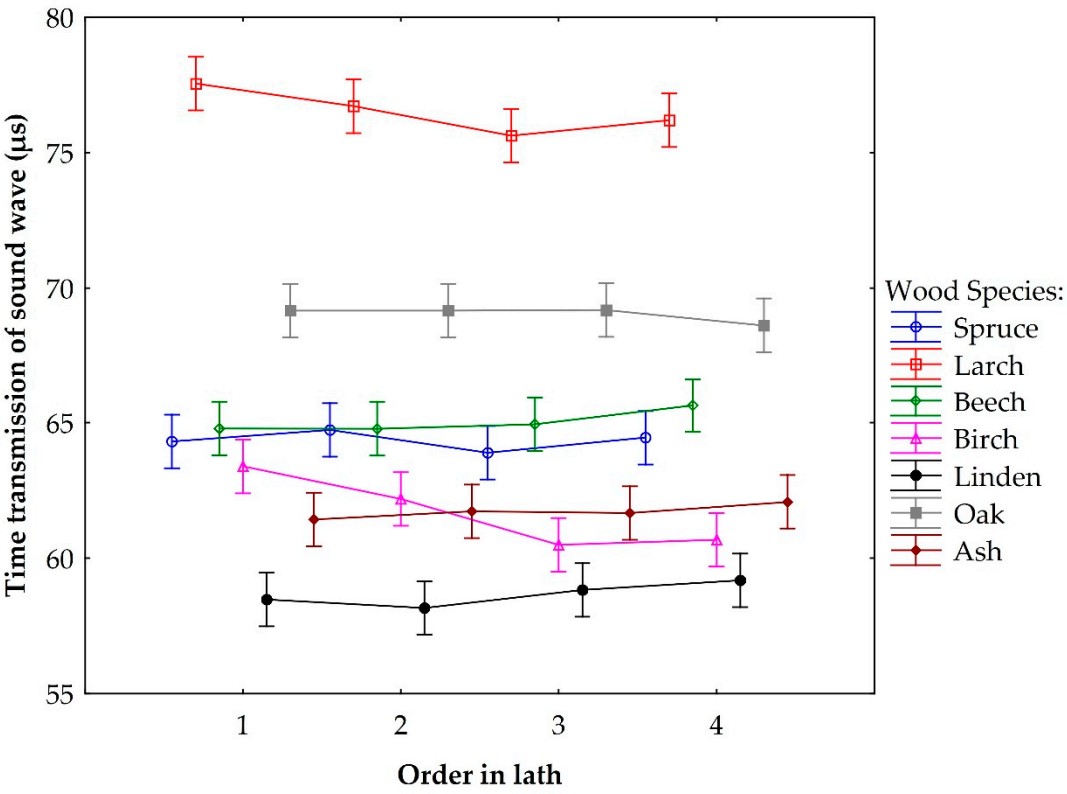

**Figure 6.** Graphical visualization of the transmission time of a sound wave of a specific wood species and samples in the parallel (longitudinal) direction in the lath.

It is clear from the graphical representation (Figure 7) that the highest values of static bending strength were achieved by birch wood when loading test specimens in the radial direction. The mean value of the bending strength limit in this case was 112.4 MPa. The lowest mean value was achieved by spruce wood under load in the radial direction, i.e., 87.1 MPa. Larch, beech and ash wood showed similar average values of static bending strength in both loading directions, i.e., in the range of 104.6 to 111.6 MPa. The mean value of bending strength for oak wood was 94.3 MPa in the tangential direction and 101.7 MPa in the radial direction of the load. All of the detected values of static bending

strength for the selected wood species under load in the radial or tangential direction are given above in Tables 1–7. It is clear from the above tables that the highest degree of variability in the values of the bending strength limit was shown by oak wood, and the lowest degree of variability was observed in ash wood, in both directions. Figure 7 also shows that the largest differences in the mean values of bending strength between the directions were achieved by oak wood, whilst lower differences were observed for linden and birch wood. Spruce, larch, beech and ash wood showed almost no differences in static bending strength values depending on the direction of the test specimen load.

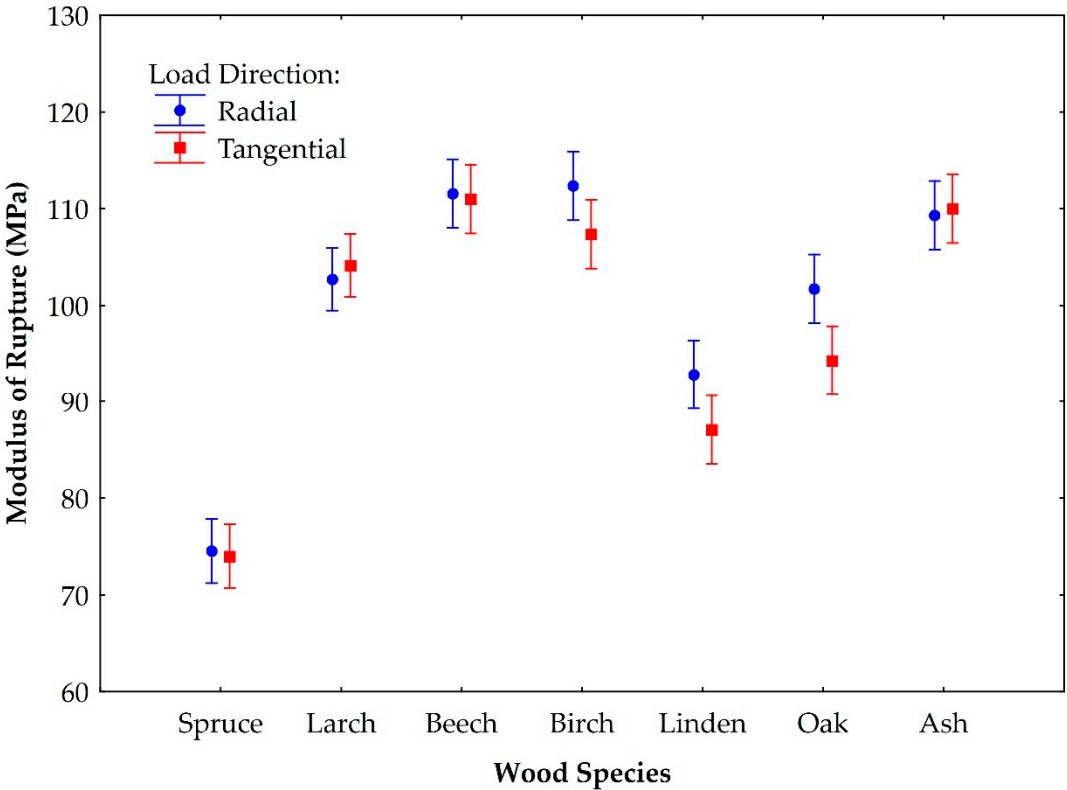

**Figure 7.** Graphic visualization of the effect of load direction on modulus of rupture (bending strength) for specific wood species.

Some authors state [6,13] that the ultimate strength of tangential bending can be 10% to 12% higher than that of radial bending in the wood of coniferous trees (softwoods). Požgaj et al. [6] state in their work that for deciduous wood species (hardwoods), these differences in static bending are negligible in the range of 2% to 4%.

Table 8 shows the differences in the mean values of static bending strength for wood of selected wood species depending on the direction of loading. These differences are expressed as a percentage and are related to how the mean values of static bending strength at radial load have changed from the values found in the tangential direction. The table shows that the largest difference was recorded for oak wood, i.e., 7.9%, while minor differences were found for linden wood, 6.6%, and birch, 4.7%, and for spruce, larch, beech and ash wood, these differences are negligible. Statistically significant differences in mean values between loading directions were demonstrated for oak and linden wood, and Duncan's test was used for multiple comparisons (see Appendix A Table A1).

It is evident from the above that for woods of certain deciduous wood species (oak, linden), higher and statistically significant differences in mean values of the limit bending strength depending on the direction of loading were achieved than those reported by Požgaj et al. [6]. Despite this fact, these differences are considered negligible, and in their works, some authors [5,6,34] report only

the values of static bending strength, which were obtained by standardized procedures, i.e., in the tangential direction.

The determined values of static bending strength under load in the tangential direction (Tables 1–7) for woods of selected wood species coincide with the range of values specified in [5,6,34], except for Linden wood, for which the mean value of the strength limit in static bending was significantly higher. Similarly, Pelit et al. [35] state in their work that their ascertained value of static bending strength for linden wood is 60.9 MPa, which is also significantly lower than the value determined in our research.

Tables 1–7 further show that the highest values of static modulus of elasticity were achieved by birch wood when loading test specimens in the radial direction. The mean value of the static modulus of elasticity in this case was 11,643 MPa. Linden wood achieved the lowest mean value under load in the tangential direction, i.e., 9332 MPa. Other mean values of the static modulus of elasticity depending on the direction of load in the investigated wood species obtained values between the maximum and minimum mean value given above. The highest degree of variability in the values of static modulus of elasticity was shown by oak wood, whilst the lowest degree of variability was observed in ash wood. In both cases, this variability was demonstrated in both the radial and tangential directions of loading. The determined values of the static modulus of elasticity had a similar trend depending on the direction of loading as in bending strength, except for ash wood, where the opposite trend was observed. The static modulus of elasticity values in the woods of the studied wood species are similar to those reported by some authors in their works [1,3,6]. The only exception is birch wood, for which lower values were found compared to the values reported by these authors.

For comparison, dynamic modules of elasticity are also specified in Tables 1–7. It turned out that in all of the methods, the highest mean values of dynamic modulus of elasticity were achieved by birch wood, i.e., in the range of 16,883 to 18,808 MPa. The lowest mean values were achieved by spruce wood in all of the methods, i.e., in the range of 7748 to 10,367 MPa. For larch wood, the greatest degree of variability was demonstrated using the ultrasound method (from the fronts), and for the remaining two methods, oak wood had the greatest degree of variability. Ash wood generally showed the lowest degree of variability. Furthermore, it is also evident that the highest mean values of the dynamic modulus of elasticity were achieved in almost all of the examined wood species using the ultrasound method, where piezoelectric sensors were applied to the sample surfaces, except for larch wood, where the opposite trend prevailed. The lowest mean values were achieved with the second ultrasound method (sensors were applied to the end faces of the samples), except for beech and larch wood. Dynamic modules determined via the resonance methods were slightly higher, except for spruce, larch and beech wood, where the opposite trend prevailed. In their work, Oberhofnerová et al. [21] determined the dynamic modulus of elasticity for spruce and oak wood via the ultrasound method (piezoelectric sensors were applied to the test specimen surfaces) and the resonance method, and there were similar differences between the mean values of the dynamic modulus of elasticity in oak wood as in this work. However, for spruce wood, such high differences were not recorded depending on the method used, as is also the case in this work. Holeček et al. [20] also found small differences in spruce wood depending on the method used. Based on these facts, it can be concluded that deciduous wood species have larger differences in the values of the dynamic modulus of elasticity depending on the method used than in coniferous trees. This can be explained by the fact that coniferous trees have a simpler anatomical structure than deciduous woods, and the propagation of the sound wave is to some extent influenced by this anatomical structure.

Compared to static modules of elasticity, dynamic modules of elasticity are largely overestimated. In the case of the examined coniferous tree wood, these differences were monitored up to 20%. For wood species with a scattered porous wood structure, these differences were more pronounced, i.e., in the range of 36% to 68%, and for wood species with a circularly porous structure in the range of 21% to 44%.

The graphical representation (Figure 8) shows that the highest values of impact (dynamic) strength were achieved by birch wood when loading test specimens in the tangential direction. The mean value

of impact strength in this case was 10.0 J/cm$^2$. The lowest mean value was achieved by oak wood in the tangential direction i.e., 4.5 J/cm$^2$. For linden wood, almost no difference was observed in the average values of impact strength depending on the direction of loading, with a mean value of 6.5 J/cm$^2$. The largest differences in the average values of dynamic toughness were recorded for spruce wood, which were 9.8 J/cm$^2$ in the radial direction and 6.5 J/cm$^2$ in the loading tangential direction. All of the ascertained values are specified in Tables 1–7. It is clear from the above tables that the highest degree of variability in the values of dynamic toughness was shown by spruce wood under load in the radial direction and the lowest degree of variability was observed in ash wood, i.e., both in the loading tangential and radial direction. It was also found that the mean values of impact strength depend on both the type of wood species and the direction of loading of the test specimens. There were significant differences depending on the direction of loading in spruce wood, larch, beech, oak, birch and ash. In their work, Požgaj et al. [6] state that for wood with a significant difference between spring and summer wood, i.e., for coniferous and deciduous wood species with a circularly porous structure, the work consumed for rupturing the test specimen (impact strength) is higher, in the range of 25% to 50%, in the radial than in the tangential direction of loading. Požgaj et al. [6] did not notice any significant differences in the wood of deciduous wood species with a scattered porous wood structure. It is clear from Table 8 that this claim agrees for coniferous tree wood and wood species with a circularly porous wood structure, but this is not the case for scattered porous wood species. The table shows that the largest difference in mean values was recorded for spruce wood, i.e., 50.3%, and slightly smaller differences were observed for larch wood 41.2%. Smaller differences of about 20% were recorded for beech, ash and oak wood. For birch wood, a difference was recorded with the opposite trend than for the woods mentioned above, i.e., −9.5%. This peculiarity can be explained by the fact that despite careful selection of the material and verification of its relative "homogeneity", it was found after image analysis that birch wood shows a significant, but slight ripple in the annual ring that cannot be seen with the human eye (Figure 5d). It can be concluded from this that this ripple greatly affected the impact strength values. Linden wood showed almost no difference, and this difference (−0.8%) can be considered negligible. Duncan's test (Table A2) was used for multiple comparisons. Statistically significant differences in the mean values of impact strength depending on the direction of loading were demonstrated in spruce, larch, beech, birch, oak and ash wood, and in linden wood, no statistically significant difference was demonstrated.

Overall, it can be concluded that the impact strength values between the individual studied wood species and directions of loading are highly influenced by the anatomical structure of the wood of individual wood species and the percentage representation of basic building elements (vessels, tracheids, parenchymal cells, libriform fibers, etc.). Furthermore, it can be concluded that the anatomical structure and the percentage of individual building elements are more pronounced in the determination of dynamic toughness and contribute to the difference in values in the radial and tangential directions of loading than in the case of bending strength under static loading. The values will also be influenced both by the submicroscopic structure of wood (fibrillar structure) and by the representation of basic building biopolymers of wood (cellulose, hemicelluloses, lignin) in individual anatomical elements of wood. In their work, Conrad el. al. [36] addressed the results of several authors and came to the conclusion that in the case of wood failure, there is a different crack propagation at the microscopic and submicroscopic level of wood depending on the direction of loading, from which it can be deduced that different crack propagation in wood from various selected wood species will also affect the resulting dynamic toughness values. The same can be assumed for static loads in bending (see Figures 9 and 10). Furthermore, it can be concluded that with dynamic loading of wood in the radial direction, there is a higher resistance of the material, because the impact energy must alternately pass through the spring and summer wood, which does not occur with loading in the tangential direction.

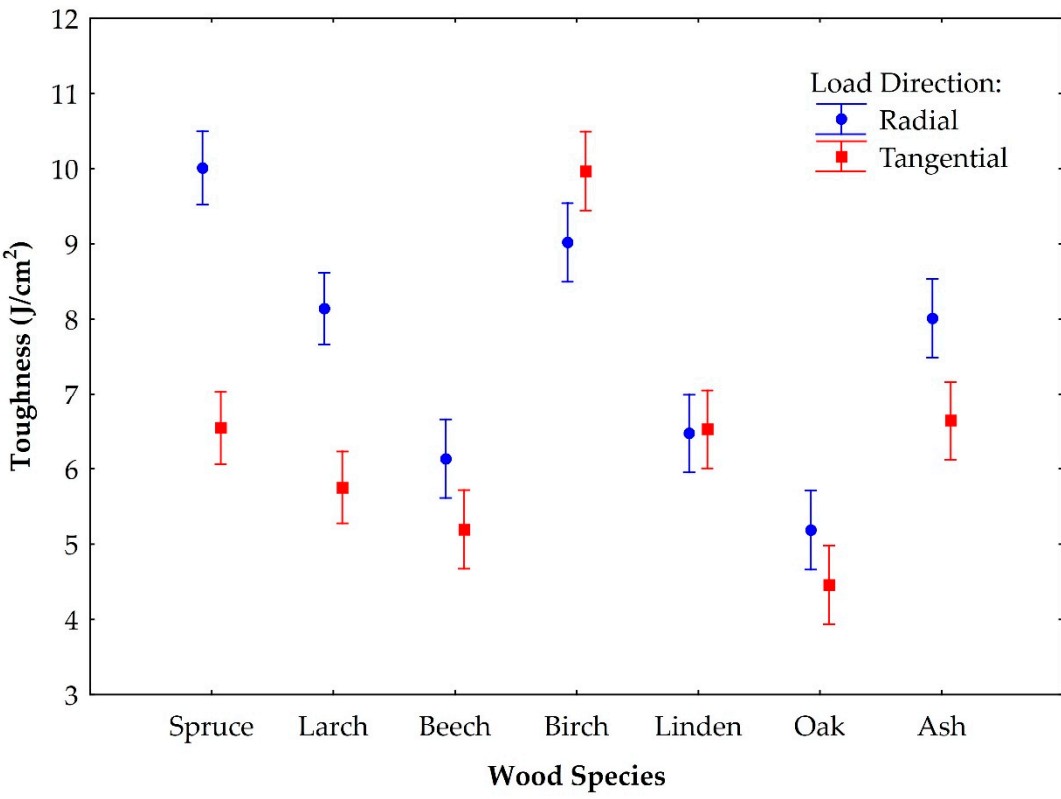

**Figure 8.** Graphic visualization of the effect of load direction on toughness (impact bending strength) for specific wood species.

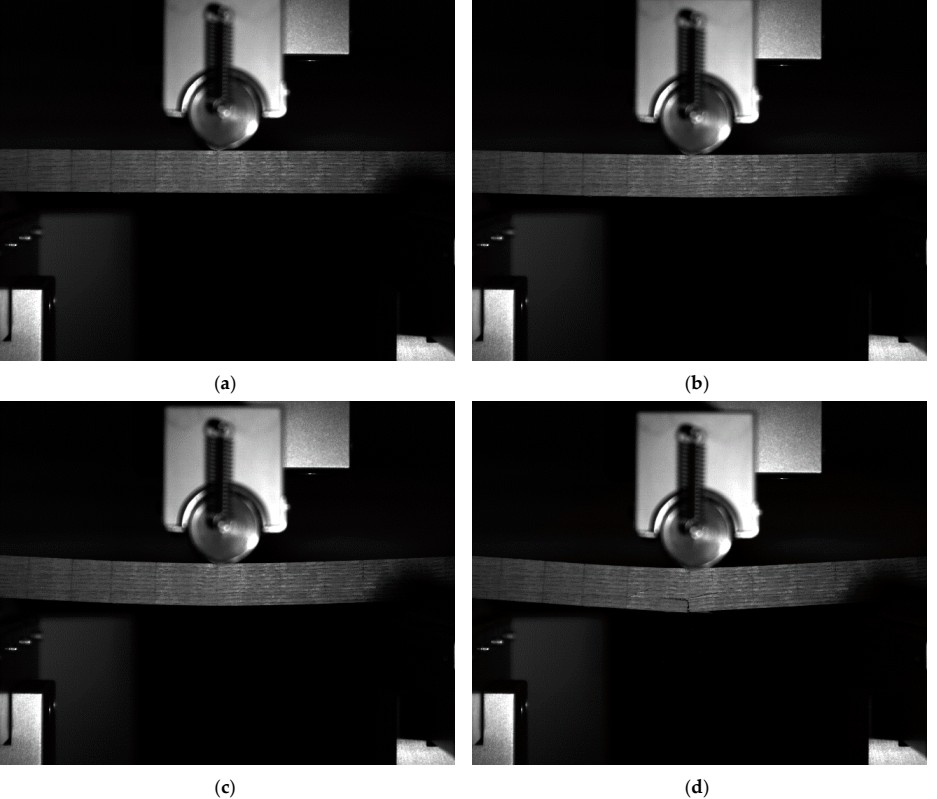

**Figure 9.** Time course of bending strength test for oak wood: 0 s (**a**), 30 s (**b**), 60 s (**c**) and 90 s (**d**).

It is clear that various methodologies can be used to determine the bending characteristics of wood, often using nondestructive principles, such as Near-Infrared Spectroscopy [37]. However, the standardized methodologies can be said to be, from a principle point of view and in compliance with the conditions of the experiments, still the most accurate methods, especially from the point of view of the validity of the obtained results.

With regard to the investigated wood species, it is often possible to observe dependencies between the individual obtained quantities or properties. Based on this fact, correlation matrices were created (see Tables A4–A10), by means of which the obtained quantities of wood of individual wood species were compared and, if possible, also depending on the direction of loading. As an example, some of the correlations were converted to graphical form (linear regression, see Figure 11). From the above graphs and tables, it is clear that relatively strong dependencies were observed between some variables, i.e., in particular in the case of oak wood, where dependence was proven in almost all cases. Dependence was not proven between impact toughness (tangential direction) and bending strength (radial direction), which is of course one of the dependencies without any significant justification. Furthermore, the dependence between impact strength and the dynamic modulus of elasticity was not demonstrated for oak wood. In this case, a statistically significant dependence was demonstrated only between the impact strength (radial direction) and the dynamic modulus of elasticity (radial direction) with a correlation coefficient of r = 0.314. In all other cases, statistically significant dependencies were demonstrated, with higher correlation coefficients than in the previous case. It is evident from the graphical expression (Figure 11a,b) that a medium–strong dependence (r ≅ 0.50 ÷ 0.75) was demonstrated between bending strength and impact strength, as well as between impact strength and static modulus of elasticity (Figure 11d). Similar statistically significant dependencies were also observed in oak wood in other cases. Very strong dependences (r > 0.75) were demonstrated between bending strength and static modulus of elasticity (Figure 11c), between bending strength and density (Figure 11f) and between the static modulus of elasticity and density (Figure 11g), as shown, for example, by Dinwoodie [2]. All of the other investigated dependencies, both for oak and the wood of other investigated wood species, are specified in Tables A4–A10.

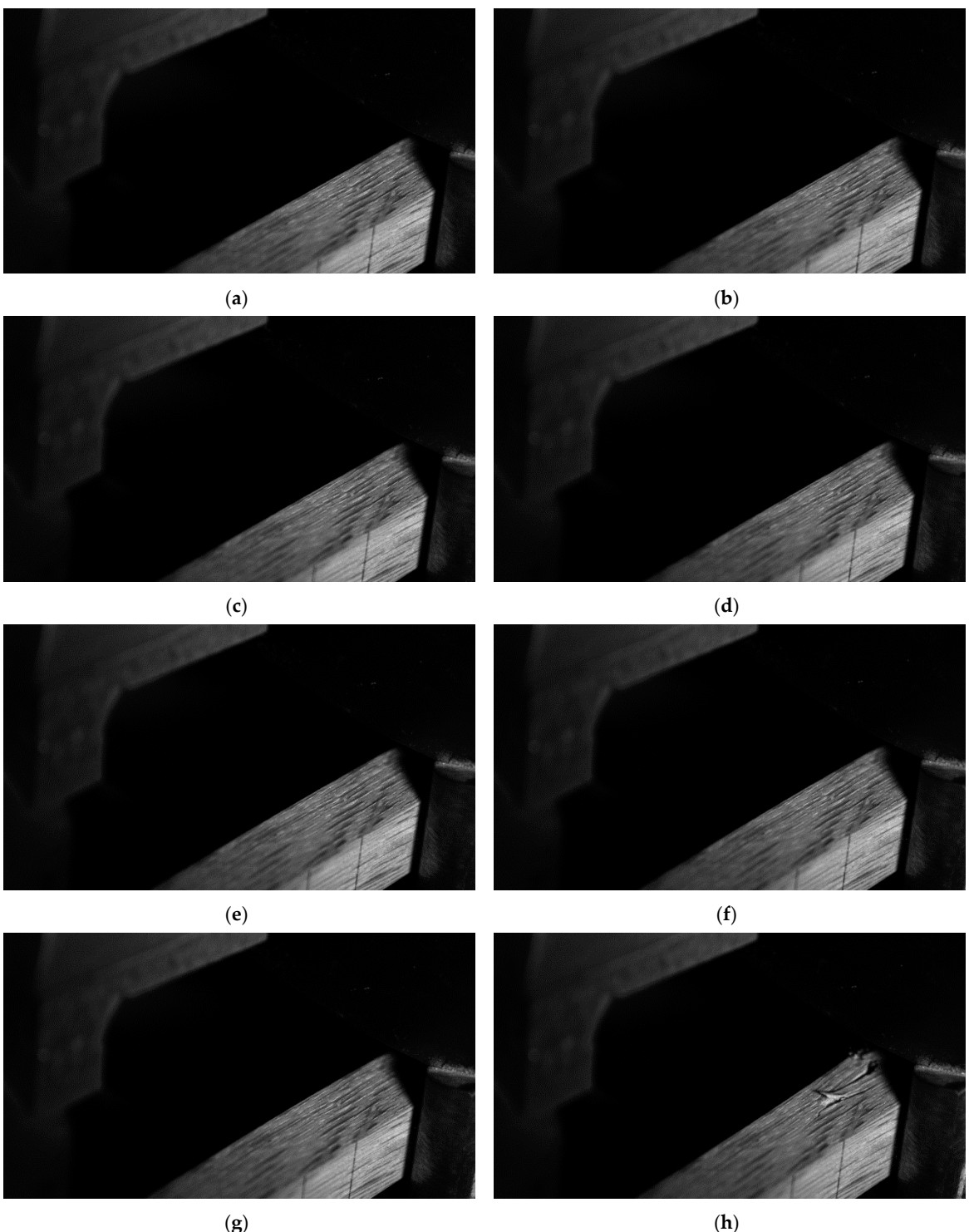

**Figure 10.** Time course of impact bending strength test for oak wood: 0 ms (**a**), 0.31 ms (**b**), 0.62 ms (**c**), 0.93 ms (**d**), 1.24 ms (**e**), 1.55 ms (**f**), 1.86 ms (**g**) and 2.17 ms (**h**).

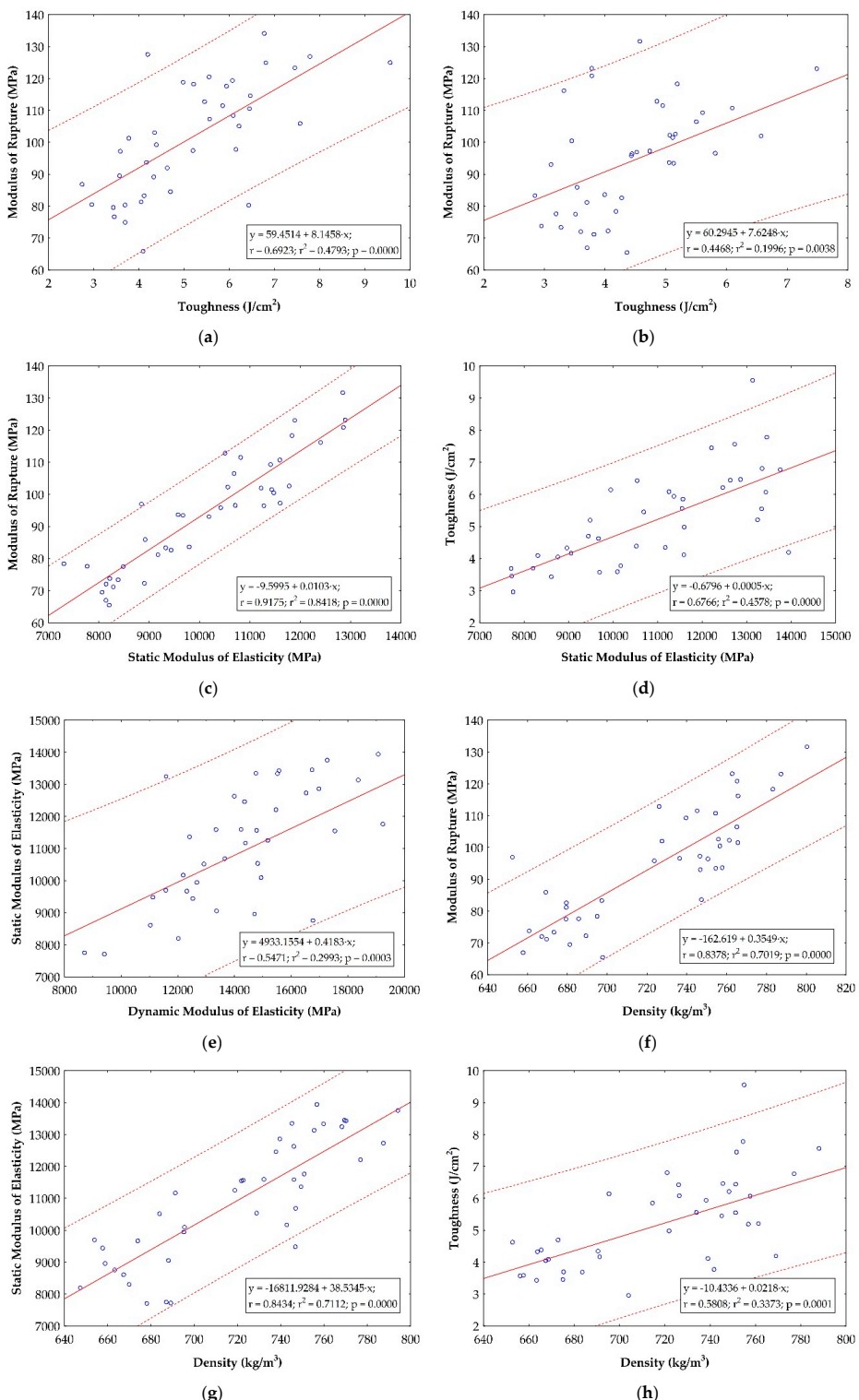

**Figure 11.** For oak wood is shown the relationship between: modulus of rupture and toughness in radial direction (**a**), modulus of rupture and toughness in tangential direction (**b**), modulus of rupture and static modulus of elasticity in tangential direction (**c**), toughness and static modulus of elasticity in radial direction (**d**), static modulus of elasticity and dynamic modulus of elasticity in radial direction (**e**), modulus of rupture and density in tangential direction (**f**), static modulus of elasticity and density in radial direction (**g**), toughness and density in radial direction (**h**).

## 4. Conclusions

The following are the most important findings of this research focused on static and dynamic bending characteristics:

1.  The largest difference of mean values in impact bending strength in the radial direction toward tangential was recorded for spruce wood, i.e., 50.3%. Slightly smaller differences were observed for larch wood, i.e., 41.2%. Smaller differences of about 20% were recorded for beech, ash and oak wood. For birch wood, a difference was recorded with the opposite trend to the woods mentioned above, i.e., −9.5%. Linden wood showed almost no difference (−0.8%). The highest values of impact strength were achieved by birch wood when loading the test specimens in the tangential direction and spruce wood in the radial direction (approximately 10 J/cm$^2$). The lowest values were achieved by oak wood in the tangential direction, on average 4.5 J/cm$^2$, and only slightly higher values in the radial direction.
2.  For static bending strength, it was found that the largest difference (radial/tangential) was recorded for oak wood, i.e., 7.9%, while minor differences were found for linden wood of 6.6% and birch of 4.7%. For spruce, larch, beech and ash wood, these differences are negligible. The highest values of static bending strength were achieved by birch wood when loading the test specimens in the radial direction (112.4 MPa). The lowest values were achieved by spruce wood under load in the radial direction—on average 87.1 MPa.
3.  Compared to static modules of elasticity, dynamic modules of elasticity are largely overestimated. In the case of the examined coniferous trees, these differences were up to a maximum of 20%. For wood species with a diffuse-porous wood structure, these differences were more pronounced, i.e., in the range of 36% to 68%, and for wood species with a ring-porous structure in the range of 21% to 43%. Regardless of the methods, in real terms, birch wood achieved the highest values of modulus of elasticity, whilst ash wood was slightly lower. The lowest values of modulus of elasticity were shown by spruce wood.
4.  It is evident from the correlation dependences between the individual quantities that most of them are statistically significant, especially in the case of oak wood, where dependence was proven in almost all cases. Very strong dependences were demonstrated between bending strength and static modulus of elasticity, between bending strength and density, as well as between static modulus of elasticity and density. It is also evident that a moderate dependence was demonstrated between bending strength and impact strength, as well as between impact strength and static modulus of elasticity, which are dependencies that are not normally monitored.

This paper is primarily intended to lead to a targeted expansion of the database dealing with bending characteristics, and thus help to understand the static and dynamic bending strength depending on the direction of the effects of external forces. Wood is very often used in structural elements of buildings; wood products (e.g., furniture), in which there is both a static load; and in many cases a dynamic load, while the direction of loading is usually not considered.

**Author Contributions:** Conceptualization, V.B.; data curation, D.N.; formal analysis, V.B. and D.N.; funding acquisition, V.B.; investigation, V.B. and D.N.; methodology, V.B.; project administration, V.B.; resources, V.B.; software, V.B. and D.N.; supervision, V.B.; validation, V.B.; visualization, V.B. and D.N.; writing—original draft, V.B., D.N. and P.Š.; writing—review and editing, V.B., D.N. and P.Š. All authors have read and agreed to the published version of the manuscript.

**Funding:** This research was funded by the Technology Agency of the Czech Republic (project No. FW01010627 Development of oak harwood bonding systems for structural and unstructural exterior applications).

**Acknowledgments:** We would like to express our thanks to the Faculty of Forestry and Wood Sciences of the Czech University of Life Sciences Prague and School Forest Enterprise of the Czech University of Life Sciences in Kostelec nad Černými Lesy.

**Conflicts of Interest:** The authors declare no conflict of interest.

# Appendix A

**Table A1.** Duncan's Multiple Range Test for Modulus of Rupture.

| MS = 130.62 DF = 546 | | Sp Rd | Sp Tg | La Rd | La Tg | Be Rd | Be Tg | Bi Rd | Bi Tg | Li Rd | Li Tg | Oa Rd | Oa Tg | As Rd | As Tg |
|---|---|---|---|---|---|---|---|---|---|---|---|---|---|---|---|
| Sp | Rd | | | | | | | | | | | | | | |
| Sp | Tg | 0.829 | | | | | | | | | | | | | |
| La | Rd | 0.000 * | 0.000 * | | | | | | | | | | | | |
| La | Tg | 0.000 * | 0.000 * | 0.566 | | | | | | | | | | | |
| Be | Rd | 0.000 * | 0.000 * | 0.001 * | 0.007 * | | | | | | | | | | |
| Be | Tg | 0.000 * | 0.000 * | 0.002 * | 0.012 * | 0.823 | | | | | | | | | |
| Bi | Rd | 0.000 * | 0.000 * | 0.000 * | 0.003 * | 0.755 | 0.618 | | | | | | | | |
| Bi | Tg | 0.000 * | 0.000 * | 0.078 | 0.199 | 0.136 | 0.186 | 0.081 | | | | | | | |
| Li | Rd | 0.000 * | 0.000 * | 0.000 * | 0.000 * | 0.000 * | 0.000 * | 0.000 * | 0.000 * | | | | | | |
| Li | Tg | 0.000 * | 0.000 * | 0.000 * | 0.000 * | 0.000 * | 0.000 * | 0.000 * | 0.000 * | 0.021 * | | | | | |
| Oa | Rd | 0.000 * | 0.000 * | 0.688 | 0.361 | 0.000 * | 0.001 * | 0.000 * | 0.037 * | 0.001 * | 0.000 * | | | | |
| Oa | Tg | 0.000 * | 0.000 * | 0.001 * | 0.000 * | 0.000 * | 0.000 * | 0.000 * | 0.000 * | 0.558 | 0.005 * | 0.003 * | | | |
| As | Rd | 0.000 * | 0.000 * | 0.014 * | 0.049 * | 0.421 | 0.530 | 0.290 | 0.430 | 0.000 * | 0.000 * | 0.005 * | 0.000 * | | |
| As | Tg | 0.000 * | 0.000 * | 0.007 * | 0.030 * | 0.558 | 0.686 | 0.399 | 0.322 | 0.000 * | 0.000 * | 0.002 * | 0.000 * | 0.788 | |

* Values are significant at $p < 0.05$. Error: between MS = mean squares, DF = degrees of freedom. Wood Species: Sp = Spruce; La = Larch; Be = Beech; Bi = Birch; Li = Linden; Oa = Oak; As = Ash. Load Direction: Rd = Radial; Tg = Tangential.

**Table A2.** Duncan's Multiple Range Test for Toughness.

| MS = 2.8329 DF = 546 | | Sp Rd | Sp Tg | La Rd | La Tg | Be Rd | Be Tg | Bi Rd | Bi Tg | Li Rd | Li Tg | Oa Rd | Oa Tg | As Rd | As Tg |
|---|---|---|---|---|---|---|---|---|---|---|---|---|---|---|---|
| Sp | Rd | | | | | | | | | | | | | | |
| Sp | Tg | 0.000 * | | | | | | | | | | | | | |
| La | Rd | 0.000 * | 0.000 | | | | | | | | | | | | |
| La | Tg | 0.000 * | 0.053 | 0.000 * | | | | | | | | | | | |
| Be | Rd | 0.000 * | 0.311 | 0.000 * | 0.301 | | | | | | | | | | |
| Be | Tg | 0.000 * | 0.001 * | 0.000 * | 0.130 | 0.015 * | | | | | | | | | |
| Bi | Rd | 0.010 * | 0.000 * | 0.017 * | 0.000 * | 0.000 * | 0.000 * | | | | | | | | |
| Bi | Tg | 0.910 | 0.000 * | 0.000 * | 0.000 * | 0.000 * | 0.000 * | 0.010 * | | | | | | | |
| Li | Rd | 0.000 * | 0.852 | 0.000 * | 0.063 | 0.355 | 0.001 * | 0.000 * | 0.000 * | | | | | | |
| Li | Tg | 0.000 * | 0.959 | 0.000 * | 0.053 | 0.315 | 0.001 * | 0.000 * | 0.000 * | 0.882 | | | | | |
| Oa | Rd | 0.000 * | 0.001 * | 0.000 * | 0.147 | 0.017 * | 0.982 | 0.000 * | 0.000 * | 0.001 * | 0.001 * | | | | |
| Oa | Tg | 0.000 * | 0.000 * | 0.000 * | 0.001 * | 0.000 * | 0.058 | 0.000 * | 0.000 * | 0.000 * | 0.000 * | 0.048 * | | | |
| As | Rd | 0.000 * | 0.000 * | 0.729 | 0.000 * | 0.000 * | 0.000 * | 0.009 * | 0.000 * | 0.000 * | 0.000 * | 0.000 * | 0.000 * | | |
| As | Tg | 0.000 * | 0.793 | 0.000 * | 0.032 * | 0.223 | 0.000 * | 0.000 * | 0.000 * | 0.680 | 0.770 | 0.000 * | 0.000 * | 0.000 * | |

* Values are significant at $p < 0.05$. Error: between MS = mean squares, DF = degrees of freedom. Wood Species: Sp = Spruce; La = Larch; Be = Beech; Bi = Birch; Li = Linden; Oa = Oak; As = Ash. Load Direction: Rd = Radial; Tg = Tangential.

**Table A3.** Duncan´s Multiple Range Test for Density.

| MS = 1792.1 DF = 1106 | | Sp Rd | Sp Tg | La Rd | La Tg | Be Rd | Be Tg | Bi Rd | Bi Tg | Li Rd | Li Tg | Oa Rd | Oa Tg | As Rd | As Tg |
|---|---|---|---|---|---|---|---|---|---|---|---|---|---|---|---|
| Sp | Rd | | | | | | | | | | | | | | |
| Sp | Tg | 0.976 | | | | | | | | | | | | | |
| La | Rd | 0.000 * | 0.000 * | | | | | | | | | | | | |
| La | Tg | 0.000 * | 0.000 * | 0.031 * | | | | | | | | | | | |
| Be | Rd | 0.000 * | 0.000 * | 0.180 | 0.001 * | | | | | | | | | | |
| Be | Tg | 0.000 * | 0.000 * | 0.415 | 0.004 * | 0.541 | | | | | | | | | |
| Bi | Rd | 0.000 * | 0.000 * | 0.006 * | 0.000 * | 0.153 | 0.050 | | | | | | | | |
| Bi | Tg | 0.000 * | 0.000 * | 0.003 * | 0.000 * | 0.107 | 0.031 * | 0.812 | | | | | | | |
| Li | Rd | 0.000 * | 0.000 * | 0.000 * | 0.000 * | 0.000 * | 0.000 * | 0.000 * | 0.000 * | | | | | | |
| Li | Tg | 0.000 * | 0.000 * | 0.000 * | 0.000 * | 0.000 * | 0.000 * | 0.000 * | 0.000 * | 0.981 | | | | | |
| Oa | Rd | 0.000 * | 0.000 * | 0.020 * | 0.000 * | 0.289 | 0.113 | 0.656 | 0.523 | 0.000 * | 0.000 * | | | | |
| Oa | Tg | 0.000 * | 0.000 * | 0.057 | 0.000 * | 0.518 | 0.238 | 0.385 | 0.293 | 0.000 * | 0.000 * | 0.628 | | | |
| As | Rd | 0.000 * | 0.000 * | 0.000 * | 0.000 * | 0.000 * | 0.000 * | 0.000 * | 0.000 * | 0.000 * | 0.000 * | 0.000 * | 0.000 * | | |
| As | Tg | 0.000 * | 0.000 * | 0.000 * | 0.000 * | 0.000 * | 0.000 * | 0.000 * | 0.000 * | 0.000 * | 0.000 * | 0.000 * | 0.000 * | 0.864 | |

* Values are significant at $p < 0.05$. Error: between MS = mean squares, DF = degrees of freedom. Wood Species: Sp = Spruce; La = Larch; Be = Beech; Bi = Birch; Li = Linden; Oa = Oak; As = Ash. Load Direction: Rd = Radial; Tg = Tangential.

**Table A4.** Values of Correlation Coefficients Between Given Quantities for Spruce Wood.

| | | MOR | | Toughness | | Static MOE | | Dynamic MOE * | | Density | |
|---|---|---|---|---|---|---|---|---|---|---|---|
| | | Radial | Tangential | Radial | Tangential | Radial | Tangential | Radial | Tangential | Radial | Tangential |
| MOR | Radial | | | | | | | | | | |
| | Tangential | 0.092 | | | | | | | | | |
| Toughness | Radial | 0.429 | −0.015 | | | | | | | | |
| | Tangential | 0.043 | 0.234 | 0.479 | | | | | | | |
| Static MOE | Radial | 0.738 | 0.177 | 0.370 | 0.154 | | | | | | |
| | Tangential | 0.548 | 0.712 | 0.300 | 0.237 | 0.508 | | | | | |
| Dynamic MOE * | Radial | 0.399 | 0.013 | −0.050 | 0.294 | 0.470 | 0.371 | | | | |
| | Tangential | 0.420 | 0.209 | 0.227 | 0.190 | 0.367 | 0.522 | 0.466 | | | |
| Density | Radial | 0.645 | 0.411 | 0.377 | 0.295 | 0.624 | 0.736 | 0.444 | 0.617 | | |
| | Tangential | 0.618 | 0.342 | 0.135 | 0.022 | 0.578 | 0.632 | 0.249 | 0.548 | 0.879 | |

Red coloring = Significant at $p < 0.05$; MOR = Modulus of Rupture; MOE = Dynamic Modulus of Elasticity; * determined by the Ultrasound Method (from the side surface).

**Table A5.** Values of Correlation Coefficients Between Given Quantities for Larch Wood.

| | | MOR | | Toughness | | Static MOE | | Dynamic MOE * | | Density | |
|---|---|---|---|---|---|---|---|---|---|---|---|
| | | Radial | Tangential | Radial | Tangential | Radial | Tangential | Radial | Tangential | Radial | Tangential |
| MOR | Radial | | | | | | | | | | |
| | Tangential | 0.471 | | | | | | | | | |
| Toughness | Radial | 0.368 | 0.742 | | | | | | | | |
| | Tangential | 0.170 | 0.560 | 0.726 | | | | | | | |
| Static MOE | Radial | 0.657 | 0.561 | 0.429 | 0.214 | | | | | | |
| | Tangential | 0.394 | 0.833 | 0.689 | 0.500 | 0.770 | | | | | |
| Dynamic MOE * | Radial | 0.424 | 0.355 | 0.727 | 0.540 | 0.719 | 0.597 | | | | |
| | Tangential | 0.308 | 0.697 | 0.507 | 0.408 | 0.633 | 0.781 | 0.427 | | | |
| Density | Radial | 0.475 | 0.516 | 0.337 | 0.175 | 0.417 | 0.361 | 0.082 | 0.385 | | |
| | Tangential | 0.428 | 0.623 | 0.319 | 0.183 | 0.355 | 0.410 | 0.001 | 0.425 | 0.912 | |

Red coloring = Significant at $p < 0.05$; MOR = Modulus of Rupture; MOE = Dynamic Modulus of Elasticity; * determined by the Ultrasound Method (from the side surface).

**Table A6.** Values of Correlation Coefficients Between Given Quantities for Beech Wood.

| | | MOR | | Toughness | | Static MOE | | Dynamic MOE * | | Density | |
|---|---|---|---|---|---|---|---|---|---|---|---|
| | | Radial | Tangential | Radial | Tangential | Radial | Tangential | Radial | Tangential | Radial | Tangential |
| MOR | Radial | | | | | | | | | | |
| | Tangential | 0.670 | | | | | | | | | |
| Toughness | Radial | 0.390 | 0.319 | | | | | | | | |
| | Tangential | 0.197 | 0.155 | 0.019 | | | | | | | |
| Static MOE | Radial | 0.867 | 0.591 | 0.478 | 0.219 | | | | | | |
| | Tangential | 0.616 | 0.638 | 0.323 | 0.547 | 0.745 | | | | | |
| Dynamic MOE * | Radial | 0.586 | 0.453 | 0.312 | 0.299 | 0.482 | 0.584 | | | | |
| | Tangential | 0.147 | 0.156 | 0.152 | 0.413 | 0.134 | 0.324 | 0.322 | | | |
| Density | Radial | 0.110 | 0.204 | 0.315 | 0.066 | 0.189 | 0.192 | 0.119 | −0.051 | | |
| | Tangential | 0.498 | 0.515 | 0.221 | −0.047 | 0.508 | 0.517 | 0.435 | 0.055 | 0.799 | |

Red coloring = Significant at $p < 0.05$; MOR = Modulus of Rupture; MOE = Dynamic Modulus of Elasticity; * determined by the Ultrasound Method (from the side surface).

**Table A7.** Values of Correlation Coefficients Between Given Quantities for Birch Wood.

| | | MOR | | Toughness | | Static MOE | | Dynamic MOE * | | Density | |
|---|---|---|---|---|---|---|---|---|---|---|---|
| | | Radial | Tangential | Radial | Tangential | Radial | Tangential | Radial | Tangential | Radial | Tangential |
| MOR | Radial | | | | | | | | | | |
| | Tangential | −0.038 | | | | | | | | | |
| Toughness | Radial | −0.351 | −0.019 | | | | | | | | |
| | Tangential | −0.468 | 0.346 | 0.548 | | | | | | | |
| Static MOE | Radial | 0.874 | 0.059 | −0.483 | −0.554 | | | | | | |
| | Tangential | 0.170 | 0.773 | −0.161 | 0.138 | 0.376 | | | | | |
| Dynamic MOE * | Radial | 0.635 | 0.015 | −0.095 | −0.150 | 0.714 | 0.267 | | | | |
| | Tangential | 0.295 | 0.057 | −0.295 | −0.021 | 0.459 | 0.341 | 0.300 | | | |
| Density | Radial | 0.002 | 0.458 | 0.594 | 0.530 | 0.149 | 0.636 | 0.083 | 0.044 | | |
| | Tangential | −0.069 | 0.369 | 0.550 | 0.654 | −0.031 | 0.487 | 0.034 | −0.022 | 0.753 | |

Red coloring = Significant at $p < 0.05$; MOR = Modulus of Rupture; MOE = Dynamic Modulus of Elasticity; * determined by the Ultrasound Method (from the side surface).

**Table A8.** Values of Correlation Coefficients Between Given Quantities for Linden Wood.

| | | MOR | | Toughness | | Static MOE | | Dynamic MOE * | | Density | |
|---|---|---|---|---|---|---|---|---|---|---|---|
| | | Radial | Tangential | Radial | Tangential | Radial | Tangential | Radial | Tangential | Radial | Tangential |
| MOR | Radial | | | | | | | | | | |
| | Tangential | 0.198 | | | | | | | | | |
| Toughness | Radial | 0.093 | 0.054 | | | | | | | | |
| | Tangential | −0.061 | 0.191 | 0.128 | | | | | | | |
| Static MOE | Radial | 0.842 | 0.266 | 0.043 | 0.153 | | | | | | |
| | Tangential | 0.070 | 0.926 | −0.016 | 0.233 | 0.199 | | | | | |
| Dynamic MOE * | Radial | 0.513 | 0.451 | 0.451 | 0.265 | 0.639 | 0.505 | | | | |
| | Tangential | 0.108 | 0.546 | 0.107 | 0.120 | 0.285 | 0.646 | 0.552 | | | |
| Density | Radial | 0.429 | 0.560 | 0.191 | −0.016 | 0.494 | 0.641 | 0.541 | 0.510 | | |
| | Tangential | 0.177 | 0.646 | 0.051 | 0.057 | 0.370 | 0.763 | 0.422 | 0.498 | 0.878 | |

Red coloring = Significant at $p < 0.05$; MOR = Modulus of Rupture; MOE = Dynamic Modulus of Elasticity; * determined by the Ultrasound Method (from the side surface).

**Table A9.** Values of Correlation Coefficients Between Given Quantities for Oak Wood.

| | | MOR | | Toughness | | Static MOE | | Dynamic MOE * | | Density | |
|---|---|---|---|---|---|---|---|---|---|---|---|
| | | Radial | Tangential | Radial | Tangential | Radial | Tangential | Radial | Tangential | Radial | Tangential |
| MOR | Radial | | | | | | | | | | |
| | Tangential | 0.708 | | | | | | | | | |
| Toughness | Radial | 0.692 | 0.680 | | | | | | | | |
| | Tangential | 0.305 | 0.447 | 0.509 | | | | | | | |
| Static MOE | Radial | 0.874 | 0.754 | 0.677 | 0.342 | | | | | | |
| | Tangential | 0.812 | 0.918 | 0.726 | 0.390 | 0.916 | | | | | |
| Dynamic MOE * | Radial | 0.388 | 0.423 | 0.314 | 0.159 | 0.547 | 0.539 | | | | |
| | Tangential | 0.409 | 0.442 | 0.073 | 0.207 | 0.487 | 0.504 | 0.800 | | | |
| Density | Radial | 0.775 | 0.845 | 0.581 | 0.513 | 0.843 | 0.882 | 0.429 | 0.405 | | |
| | Tangential | 0.791 | 0.838 | 0.519 | 0.549 | 0.816 | 0.866 | 0.379 | 0.360 | 0.979 | |

Red coloring = Significant at $p < 0.05$; MOR = Modulus of Rupture; MOE = Dynamic Modulus of Elasticity; * determined by the Ultrasound Method (from the side surface).

**Table A10.** Values of Correlation Coefficients Between Given Quantities for Ash Wood.

| | | MOR | | Toughness | | Static MOE | | Dynamic MOE * | | Density | |
|---|---|---|---|---|---|---|---|---|---|---|---|
| | | Radial | Tangential | Radial | Tangential | Radial | Tangential | Radial | Tangential | Radial | Tangential |
| MOR | Radial | | | | | | | | | | |
| | Tangential | 0.388 | | | | | | | | | |
| Toughness | Radial | 0.170 | 0.032 | | | | | | | | |
| | Tangential | 0.130 | 0.054 | 0.295 | | | | | | | |
| Static MOE | Radial | 0.670 | 0.532 | 0.311 | 0.132 | | | | | | |
| | Tangential | 0.455 | 0.663 | 0.375 | 0.318 | 0.755 | | | | | |
| Dynamic MOE * | Radial | 0.022 | 0.317 | 0.371 | 0.003 | 0.345 | 0.307 | | | | |
| | Tangential | 0.029 | 0.337 | 0.088 | 0.163 | 0.455 | 0.430 | 0.334 | | | |
| Density | Radial | 0.509 | 0.550 | 0.254 | 0.339 | 0.823 | 0.734 | 0.337 | 0.389 | | |
| | Tangential | 0.602 | 0.646 | 0.169 | 0.188 | 0.845 | 0.747 | 0.265 | 0.373 | 0.938 | |

Red coloring = Significant at $p < 0.05$; MOR = Modulus of Rupture; MOE = Dynamic Modulus of Elasticity; * determined by the Ultrasound Method (from the side surface).

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
