# Peer review of "Comparison and Analysis of Radial and Tangential Bending of Softwood and Hardwood at Static and Dynamic Loading"

_forests, doi:10.3390/f11080896_

Round 1
Reviewer 1 Report
The topic of this work is quite interesting. This work involved many different tests of mechanical strength determination, examining different testing methods and 5 different wood species, extending much the database of wood species data.
As you refer in the text, aim of the study is the targeted expansion of the database referring to static and dynamic bending strength, and since we are talking about an already existent database including many works results found in the literature providing data on this field, this proves unfortunately a relative lack of novelty in this work.
Some grammatical errors were detected in the text of manuscript. Please, make an extended check for such errors. The phrase "largely influenced" could be altered to "highly" influenced, as well as the word "greater" could be changed. The introduction could be enriched with bibliography, many significant relevant researches previously carried out on the specific field that are not analyzed or taken into consideration.
You should add which was the age of the trees cut for the purposes of the specific research. You should also add the information on how many different trees/stems were used. Did you base the selection of the specimens (sampling method) on a specific standard. The materials-methods chapter is well prepared providing the necessary information, except the cases I commented earlier. The figures of the manuscript are necessary and helpful to the reader, but should be changed to be clearer and of higher lightness. The chapter of results is properly prepared. The results and tendencies are well described.
Author Response
The topic of this work is quite interesting. This work involved many different tests of mechanical strength determination, examining different testing methods and 5 different wood species, extending much the database of wood species data.
Thank you very much for your opinion on the article. The article deals with as many as seven wood species, and it also includes the determination of both elastic and mechanical characteristics. We would also like to thank you for your positive opinion on the solved topic.
As you refer in the text, aim of the study is the targeted expansion of the database referring to static and dynamic bending strength, and since we are talking about an already existent database including many works results found in the literature providing data on this field, this proves unfortunately a relative lack of novelty in this work.
The novelty lies in the extension of the database of those bending characteristics that depend on the direction of external forces, which is not so common and can be seen in the information in the first paragraph of the introduction to this paper. For this reason, it was the precise selection of the sample material that was emphasized in line with the effort to eliminate the heterogeneity of the wood, which resulted in the ideal "homogeneity" of parallel test specimens, cf. line 56-60. More or less, the only relevant information about the influence of direction in dynamic bending is mentioned in the discussion (line 432-437). As far as is static bending concerned, the databases are sufficient, but the correctly methodically mentioned "parallelism" of the test samples has not been solved yet.
Some grammatical errors were detected in the text of manuscript. Please, make an extended check for such errors. The phrase "largely influenced" could be altered to "highly" influenced, as well as the word "greater" could be changed. The introduction could be enriched with bibliography, many significant relevant researches previously carried out on the specific field that are not analyzed or taken into consideration.
Grammar mistakes have been corrected, lines 60, 287, 296, 337, 425, and 439 in the "doc" file (60, 312, 321, 365, 455, and 470 in the "pdf" file). The information about the introduction, cf. the comment in the previous paragraph.
You should add which was the age of the trees cut for the purposes of the specific research. You should also add the information on how many different trees/stems were used. Did you base the selection of the specimens (sampling method) on a specific standard. The materials-methods chapter is well prepared providing the necessary information, except the cases I commented earlier. The figures of the manuscript are necessary and helpful to the reader, but should be changed to be clearer and of higher lightness. The chapter of results is properly prepared. The results and tendencies are well described.
In response to the above-mentioned comment, it is quite clear that both the age of the trees and other data have no justification. It is essential to conduct research on quality samples from the central board of the cutout with regard to the direction of the loading force. The above-mentioned data prove that the information given in the first paragraph of Chapter 2 is sufficient. To make the information complete, there were two 2-meter sections of each wood species from one tree in the maturity age.

Reviewer 2 Report
The manuscript has been prepared carefully and is scientifically valuable. However, it requires some improvements:
- A large part of the manuscript is data, charts and figures. I suggest refining the data presentation to better guide the reader through the most important findings. For example, tables 1-7 could be combined into one. Other presentation of the data could include the results of statistical analyzes, e.g.: Duncan's test, using homogeneous groups. This would reduce the number of tables and make the paper clearer.
-
The methodology is described too detailed. It should be synthetic and enable the reproduction of the research. When refereeing to standards there is no need for extensive description. Only modifications of the methods should be labelled (e.g. the support spacing in the standard is given as 240 mm, while in the experiment it was 200 mm, etc.). This applies to the entire methodology. There is no need to provide formulas included in the standards. The current form of the methodology is unacceptable.
-
Figures 2, 3 are redundant, they add no value to the paper.
-
Photos attached to the manuscript misses metric reference - a scale enabling the interpretation of the photos should be included (e.g. Fig 5.). Mentioning the scale only in the text is inappropriate.
- Figures 9 and 10 – contribution of the mentioned in the text elements in the photo is less than 50% of the total photo area, what is confusing for the reader – photos must be edit to better expose described elements. Moreover, figure 10, is missing the scale enabling interpretation of these pictures.
Author Response
The manuscript has been prepared carefully and is scientifically valuable. However, it requires some improvements:
A large part of the manuscript is data, charts and figures. I suggest refining the data presentation to better guide the reader through the most important findings. For example, tables 1-7 could be combined into one. Other presentation of the data could include the results of statistical analyzes, e.g.: Duncan's test, using homogeneous groups. This would reduce the number of tables and make the paper clearer.
Thank you very much for your opinion on the article. The primary objective is to expand the database of bending characteristics; it means that the data in Tables 1-7 are necessary and there would be a reduction of data to mere average values, without any variability data, and other similar data. The most important comparative data can be found in Table 8 that the reader is sufficient directed to in the 1st paragraph of the results. The appendix includes several Duncan's tests; however, the solution for homogeneous groups is not optimal in this case.
The methodology is described too detailed. It should be synthetic and enable the reproduction of the research. When refereeing to standards there is no need for extensive description. Only modifications of the methods should be labelled (e.g. the support spacing in the standard is given as 240 mm, while in the experiment it was 200 mm, etc.). This applies to the entire methodology. There is no need to provide formulas included in the standards. The current form of the methodology is unacceptable.
Methodology reduction is a matter of opinion. The second opponent shares our opinion and considers it optimal in this version. The second opponent would even prefer to expand it. The reduction in the way proposed would more or less mean that it should be virtually omitted in that the number of deviations from standards or commonly used methodologies was negligible.
Figures 2, 3 are redundant, they add no value to the paper.
It is again a matter of opinion. Some of the pictures in the articles generally serve either to make the article "more attractive" or to be more illustrative for less experienced readers. Obviously, photos of the Tira test machine and those of similar equipment have not been included in the article.
Photos attached to the manuscript misses metric reference - a scale enabling the interpretation of the photos should be included (e.g. Fig 5.). Mentioning the scale only in the text is inappropriate.
As far as the photos of microscopic cross-section of wood species are concerned, when enlarged, they represent a standard procedure. In spite of the fact mentioned, the scale has been added. However, the analyses were of an additional character, and have not been discussed.
Figures 9 and 10 – contribution of the mentioned in the text elements in the photo is less than 50% of the total photo area, what is confusing for the reader – photos must be edit to better expose described elements. Moreover, figure 10, is missing the scale enabling interpretation of these pictures.
The figures given are also only illustrative, showing rather the initial formation of cracks; the scale is irrelevant from a point of view of the known test samples dimensions. Obviously, it is clear that the video from which the images were taken would be much more relevant. As it has already been mentioned, they mere illustrate the time and place of crack formation, cf. the text to the images.